# Imposing conservation properties in deep dynamics modeling via contrastive learning

## Abstract

Deep neural networks (DNN) has shown great capacity of modeling a dynamical system, but these DNN-based dynamical models usually do not obey conservation laws. To impose the learned DNN dynamical models with key physical properties such as conservation laws, this paper proposes a two-step approach to endow the invariant priors into the simulations. We first establish a contrastive learning framework to capture the system invariants along the trajectory observations. During the dynamics modeling, we design a projection layer of DNNs to preserve the system invariance. Through experiments, we show our method consistently outperforms the baseline in both coordinate error and conservation metrics and can be further extended to complex and large dynamics by leveraging autoencoder. Notably, a byproduct of our framework is the automated conservation law discovery for dynamical systems with single conservation property.

## 1 Introduction

With the quick growth of computational resources and massive prediction power of neural networks, recent times have seen great success of artificial intelligence in a wide range of applications such as image classification (He et al., 2016), natural language processing (Vaswani et al., 2017; Devlin et al., 2018) and reinforcement learning (Mnih et al., 2013). Despite the scalability and diversity of modern machine learning tasks, extracting the underlying latent mechanism from training data and deploying the knowledge toward the new occurrence have always been the heart of artificial intelligence. The idea to work with a compressed representation or prior information has been historically entangled with the development of machine learning, from earlier tools like clustering and principal component analysis (PCA) to more contemporary autoencoders or embeddings. In recent years, there is a surge in interest to discover knowledge from larger or even different domains leveraging techniques like representation learning (Bengio et al., 2013; Chen & He, 2021) and transfer learning (Weiss et al., 2016) to more general meta-learning (Rusu et al., 2018; Santoro et al., 2016) and foundation models (Bommasani et al., 2021) which are capable of handling a wide range of tasks.

Many critical discoveries in the world of physics were driven by distilling the invariants from observations. For instance, the Kepler laws were found by analyzing and fitting parameters for the astronomical observations, and the mass conservation law was first carried out by a series of experiments. However, such discovery usually requires extensive human insights and customized strategies for specific problems. This naturally raises a question, can we learn certain conservation laws from real-world data in an automated fashion? On the other hand, data-driven dynamical modeling is prone to violation of physics laws or instability issues (Greydanus et al., 2019; Kolter & Manek, 2019), since the model only statistically learns the data or system state function without knowing physics prior.

In this paper, we provide a novel contrastive perspective to find one or more distinguishing features (i.e. conservation values) of physics-based system trajectories. By comparing the latent space distance of the system state observations, we aim to learn a low-dimensional representation potentially serving as the invariant term for the system. With such inspiration, we propose **ConCerNet** consisting of two neural networks. The first network contrastively learns the trajectory invariants, the second network captures the nominal system dynamical behavior which will be corrected by the first network to preserve certain properties of the simulation in the long term prediction. The correction is implemented by projecting the dynamical neural network output on the learned conservation man-

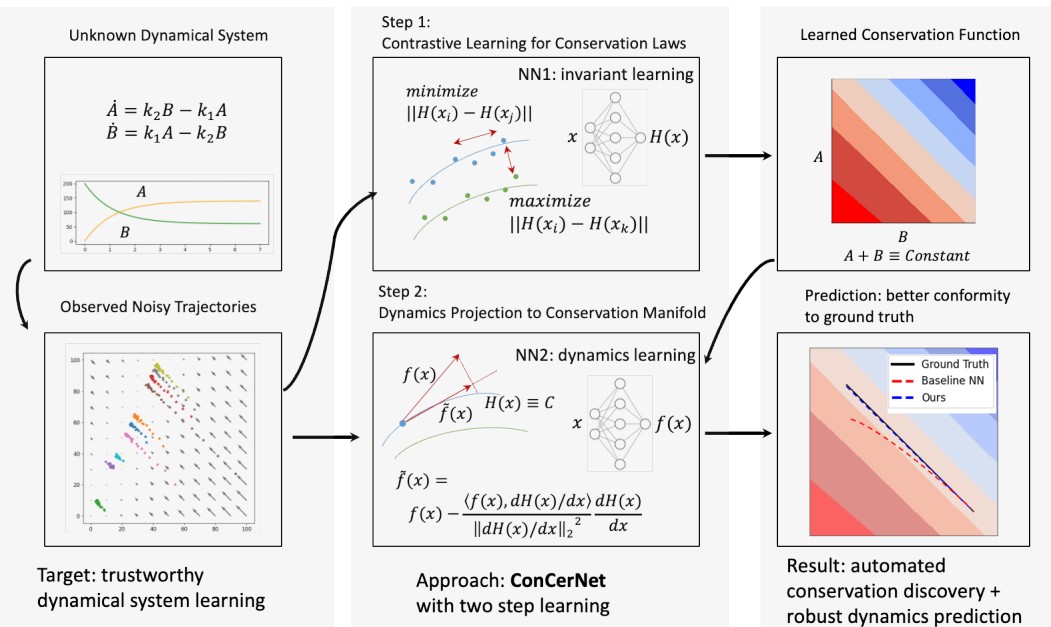

Figure 1: Pipeline to learn the dynamical system conservation and enforce it in simulation. A contrastive learning framework is proposed to extract the invariants across trajectory observations, then the dynamical model is projected to the invariant manifold to guarantee the conservation property.

ifold learned by the first module, and therefore enforcing the trajectory conservation for the learned invariants.

We summarize our main contributions as follows:

- We provide a novel contrastive learning perspective of dynamical system trajectory data to capture the invariants of dynamical systems. One byproduct of this method is that the learned invariant functions discover physical conservation laws in certain cases. To the best of the authors' knowledge, this is the first work that studies the discovery of conservation laws for general dynamical systems through contrastive learning.

- We propose a projection layer to impose any invariant function for dynamical system trajectory prediction, guaranteeing the conservation property during simulation.

- Based on the above two components, we establish a generic learning framework for dynamical system modeling named **ConCerNet** (CONtrastive ConsERved Network) which provides robustness in prediction outcomes and flexibility to be applied to a wide range of dynamical systems that mandate conservation properties. We conducted extensive experiments to demonstrate the efficacy of **ConCerNet**, especially its remarkable improvement over a generic neural network in prediction error and conservation violation metrics.

- We draw inferences on the relationship between the contrastively learned function, the exact conservation law, and the logistics of contrastive invariant learning. Further, potential improvements to the proposed method are illustrated that can improve the automated scientific discovery process.

## 2 BACKGROUND AND RELATED WORK

### 2.1 CONTRASTIVE LEARNING

Unlike discriminative models that explicitly learn the data mappings, contrastive learning aims to extract the data representation implicitly by comparing among examples. The early idea dates back to the 1990s (Bromley et al., 1993) and has been widely adopted in many areas. One related field to

our work is metric learning (Chopra et al., 2005; Harwood et al., 2017; Sohn, 2016), where the goal is to learn a distance function or latent space to cluster similar examples and separate the dis-similar ones.

Contrastive learning has been a popular choice for self-supervised learning (SSL) tasks recently, as it demonstrated its performance in many applications such as computer vision (Chen et al., 2020a; He et al., 2020; Ho & Nvasconcelos, 2020; Tian et al., 2020) and natural language processing (Wu et al., 2020b; Gao et al.). There are many existing works related to contrastive learning, covering the design of contrastive loss (Oord et al., 2018a; Chen et al., 2020a;b), memory bank (Wu et al., 2018), sample bias (Chuang et al., 2020; Arora et al., 2019) and momentum encoder (Chen et al., 2020c; He et al., 2020).

## 2.2 DEEP LEARNING BASED DYNAMICAL SYSTEM MODELING

Constructing dynamical system models from observed data is a long-standing research problem with numerous applications such as forecasting, inference and control. System identification (SYSID) (Ljung, 1998; Keesman & Keesman, 2011) was introduced a few decades ago and designed to fit the system input-output behavior with choice of lightweight basis functions. In recent years, neural networks became increasingly popular in dynamical system modeling due to its representation power. In this paper, we consider the following neural network based learning task to model an (autonomous and continuous time) dynamical system:

$$f_\theta(x) \sim \dot{x} \equiv \frac{dx(t)}{dt} \qquad (1)$$

where $x \in \mathbb{R}^n$ is the system state and $\dot{x}$ is its time derivative. $f_\theta : \mathbb{R}^n \to \mathbb{R}^n$ denotes the neural network model $f$ with parameter $\theta$ to approximate ground truth dynamics.

The vanilla neural networks learn the physics through data by minimizing the step prediction error, without a purposely designed feature to honor other metrics such as conservation laws. One path to address this issue is to include an additional loss in the training (Singh et al., 2021; Wu et al., 2020a; Richards et al., 2018; Wang et al., 2020); however, the soft Lagrangian treatment does not guarantee the model performance during testing. Imposing hard constraints upon the neural network structures is a more desirable approach, where the built-in design naturally respect certain property regardless of input data. Existing work includes: Kolter & Manek (2019) learns the dynamical system and a Lyapunov function to ensure the exponential stability of predicted system; Hamiltonian neural network (HNN, Greydanus et al. (2019)) targets at the Hamiltonian mechanics, directly learns the Hamiltonian and uses the symplectic vector field to approximate the dynamics; Lagrangian neural network (LNN, Cranmer et al. (2020)) extends the work of HNN to Lagrangian mechanics. Although the above models are able to capture certain conservation laws under specific problem formulations, they are not applicable to general conserved dynamical systems (e.g. mass conservation). This motivates our work in this paper to propose a contrastive learning framework in a more generic form that is compatible with arbitrary conservation.

## 2.3 LEARNING WITH CONSERVED PROPERTIES

Automated scientific discovery from data without prior knowledge has attracted great interest to both communities in physics and machine learning. Besides the above-mentioned HNN and LNN, a few recent works (Zhang et al., 2018; Liu & Tegmark, 2021; Ha & Jeong, 2021; Liu et al., 2022) have explored automated approaches to extract the invariants or conservation laws from data. Despite the promising results, the existing methods usually suffer from limitations including poor data sample efficiency and reliance on artificial pre-processing, and hereby difficult to extend to larger and more general systems. Note that the aim of our proposed framework is to provide a genuinely adaptable and highly automated tool for trustworthy data-driven dynamical systems modeling, rather than simply solving the above limitations. The other line of physics invariant learning focuses on discovering the hidden symmetries in physical systems (Liu & Tegmark, 2022; Mototake, 2021). To the best of our knowledge, this is the first time conservation law discovery for general dynamical systems is studied through the lens of contrastive learning.

## 3 PROPOSED METHODS

### 3.1 CONTRASTIVE LEARNING FOR CONSERVATION PROPERTY FROM SYSTEM TRAJECTORIES

In the practice of dynamical system learning, the dynamics data is usually observed as a set of trajectories of system state $\{x_t^i \in \mathbb{R}^n\}_{i=1,t=1}^{N,T}$, where $i$ denotes the trajectory index of total trajectory number $N$ and $t$ is the time step with total time step number $T$. $H_{\theta_c} : \mathbb{R}^n \to \mathbb{R}^m$ is the neural network parameterization to map the original state to latent representation with dimension $m$. We let the trajectory set $\{C_i\}_{i=1}^N$ be the simulation history starting from initial conditions drawn from a distribution $D$, and assume the initial conditions have various conservation values. Inspired by the concept of Neighborhood Component Analysis (NCA, Goldberger et al. (2004)) and Neighborhood analysis Contrastive loss (NaCl, Ko et al. (2022)), we analogize each trajectory as a neighbor class where the individual points are drawn from and try to solve a $N$ class classification problem by comparing the latent representation of every pair of points. Since the invariants are well conserved along the trajectory and differ among different trajectories, we aim to find the conservation laws which naturally serve as the latent representation of each class. We consider the nearest neighbor selection as a random event, where the probability of point $x_t^i$ belonging to the trajectory $C_k$ is defined as $p(C_k|x_t^i)$ in the following design:

$$p(C_k|x_t^i) := \frac{\sum_{t_2=1}^T \exp(-\|H_{\theta_c}(x_t^i) - H_{\theta_c}(x_{t_2}^i)\|^2)\mathbb{1}(t_2 \neq t)}{\sum_{j=1}^N \sum_{t_2=1}^T \exp(-\|H_{\theta_c}(x_t^i) - H_{\theta_c}(x_{t_2}^j)\|^2)\mathbb{1}(i \neq j \text{ or } t \neq t_2)} \tag{2}$$

By traversing all pairs of points, we formulate the contrastive loss function:

$$\mathcal{L}_{con} = \mathbb{E}_{\substack{x \sim C, C \sim D, \\ x^+ \in C_{\setminus x}, \\ x^- \in C^-, C^- \sim D_{\setminus C}}} \left[ -\log \frac{\sum_k \exp(-\|H_{\theta_c}(x) - H_{\theta_c}(x_k^+)\|^2)}{\sum_k \exp(-\|H_{\theta_c}(x) - H_{\theta_c}(x_k^+)\|^2) + \sum_k \exp(-\|H_{\theta_c}(x) - H_{\theta_c}(x_k^-)\|^2)} \right] \tag{3}$$

$$\approx \frac{1}{NT} \sum_{i=1}^N \sum_{t_1=1}^T -\log \frac{\sum_{t_2=1}^T \exp(-\|H_{\theta_c}(x_{t_1}^i) - H_{\theta_c}(x_{t_2}^i)\|^2)\mathbb{1}(t_1 \neq t_2)}{\sum_{j=1}^N \sum_{t_2=1}^T \exp(-\|H_{\theta_c}(x_{t_1}^i) - H_{\theta_c}(x_{t_2}^j)\|^2)\mathbb{1}(i \neq j \text{ or } t_1 \neq t_2)} \tag{4}$$

Similar to NCA, we choose squared Euclidean norm as the distance metric between point pairs[1] and the Softmax like function, ensuring the nice property of probabilistic distribution. To notice, in common contrastive learning setting like SimCLR (Chen et al., 2020a) or NCA, the classification target group only contains one element. In our setup, we consider the classification problem as assigning one point to a group of points in the same class, therefore we have the additional summation loop on both denominator and numerator. This setup is similar to the NaCl loss in Ko et al. (2022) with many positive pairs.

### 3.2 ENFORCING CONSERVATION LAWS IN DYNAMICAL SYSTEM PREDICTION

After the contrastive learning of the $m$ conservation terms $H_{\theta_c}(x) \in \mathbb{R}^m$, we attempt to enforce the predicted trajectory along the learned conservation manifold in the simulation stage, $s.t. \frac{dH_{\theta_c}(x)}{dt} = 0$. In the continuous dynamical system like Equation (1), we can project the nominal neural network output $f_{\theta_d}(x)$ onto the conservation manifold by eliminating its parallel component to the normal direction of the invariant planes (i.e. $\nabla_x H_{\theta_c}(x)$). We define the projected dynamical model $\tilde{f}_{\theta_d}(x)$ as following:

---

[1]Positive pair denotes correlated views of the same example in contrastive learning literature (Chen et al., 2020a), in this paper we define positive pair as two system states from the same trajectory therefore they are assumed with same conservation property and considered in the same class.

$$\tilde{f}_{\theta_d}(x) := \text{Projection}\left(f_{\theta_d}(x), \{f : \langle f, \nabla_x H_{\theta_c}(x)\rangle = 0\}\right) \tag{5}$$

$$= f_{\theta_d}(x) - \sum_{i=1}^{m} \langle f_{\theta_d}(x), (\nabla_x H^i_{\theta_c}(x))^\perp \rangle (\nabla_x H^i_{\theta_c}(x))^\perp \tag{6}$$

where $\{(\nabla_x H^i_{\theta_c}(x))^\perp\}$ denotes the orthonormalized set of vectors from $\{\nabla_x H_{\theta_c}(x)\}$ by Gram–Schmidt process. The summation symbol indicates projections to each of the conservation terms and vanishes if only 1 conservation term is learned (i.e. $m = 1$). The projected model dynamics naturally satisfies $\langle \tilde{f}_{\theta_d}(x), \nabla_x H_{\theta_c}(x)\rangle = 0$ and therefore guarantees $H_{\theta_c}(x)$ being constant along the simulated trajectory. The intuitive diagram of projection is shown in Figure 1.

For dynamical system learning, the loss function is simply the mean square loss between the neural network prediction and the observed system time derivative.

$$\mathcal{L}_{\text{dyn}} = \mathbb{E}_{x \sim C \sim D}\left[\|\tilde{f}_{\theta_d}(x) - \dot{x}\|^2\right] \tag{7}$$

## 4 EXPERIMENTS

To demonstrate the power of our method, we first illustrate the procedure through two simple conservation examples, then we show technical details to overcome a more complex problem. In the end, we highlight the method is extendable to high-dimensional problems by leveraging an autoencoder. All the system and experiment details are listed in Appendix A.

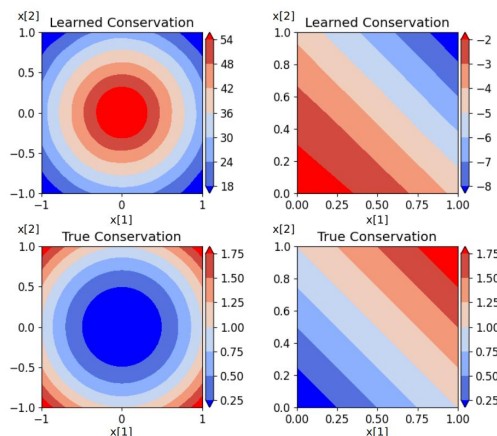

Figure 2: Learned conservation function vs ground truth, left: ideal spring mass, right: chemical kinetics

### 4.1 SIMPLE CONSERVATION EXAMPLES

In this section, we introduce two simple examples: **Ideal spring mass system** under energy conservation ($x[1]^2 + x[2]^2$) and **Chemical reaction** under mass conservation ($x[1] + x[2]$). Both systems have 2D state space for easier visualization of the learned conservation function. Figure 2 shows the learned conservation compared with ground truth. The contrastive learning process captures the quadratic and linear functions, as the contour lines are drawn in circles and affine functions. To notice, the learned conservation here is approximately the exact conservation differing by some constant coefficient, as the conservation is a relative quantify instead of an absolute value. For further relationship between contrastively learning invariants and actual conservation, we delay the discussion to Section 5.1. In Figure 3, we compare the two methods by showing the trajectory, conservation and coordinate error to the ground truth. The vanilla neural network is likely to quickly diverge from the conserved trajectory, and the error grows faster than our proposed method.

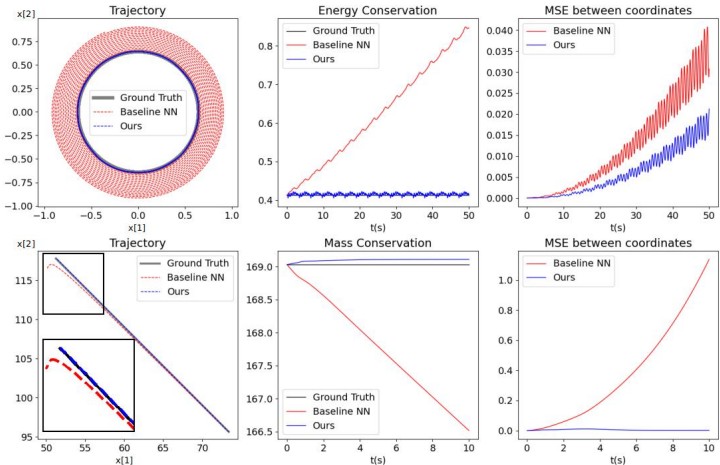

Figure 3: Simulation comparisons of two simple examples: upper row: ideal spring mass system, lower row: chemical kinematics. 1st column: state trajectories, 2nd column: violation of conservation laws to ground truth, 3rd column: mean square error to ground truth.

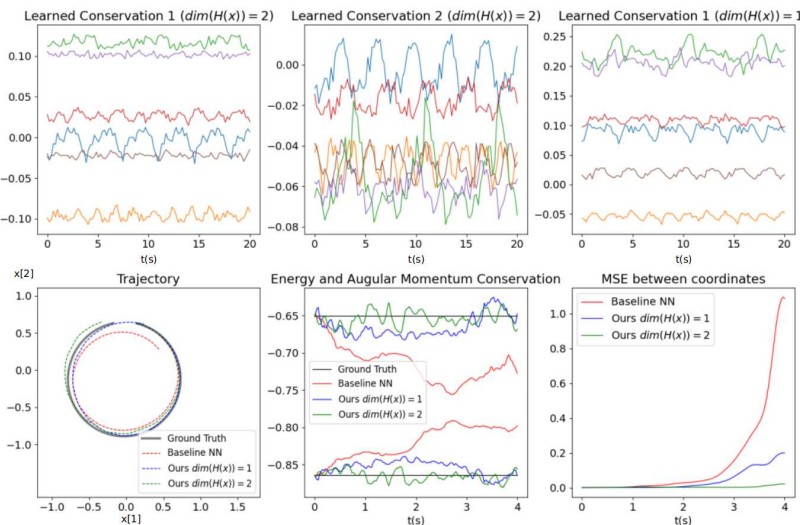

Figure 4: Kepler system results: First rows (contrastive learning) learned invariants for 6 sampled trajectories. Last row (dynamics simulation): state trajectories, violation of conservation laws to ground truth, mean square error to ground truth.

## 4.2 COMPLEX CONSERVATION FUNCTIONS

In this section, we tackle a more complex system with more than one conservation laws with complicated representations. The **Kepler system** describes a planet orbiting around a star with elliptical trajectories. The planet has four dimensional states, including both coordinates in the 2D plane and the corresponding velocity. The system has two conservation terms (energy conservation $\frac{x[3]^2+x[4]^2}{2} - \frac{1}{\sqrt{x[1]^2+x[2]^2}}$ and angular momentum conservation $x[1]x[4] - x[2]x[3]$).

The major challenge for Kepler system conservation learning is data requirement and the representation power of the simple neural network to capture the complex energy conservation function including the square root on the denominator. Besides, if we use the standard contrastive loss from Equation (3), we found it is possible that the learned conservation might converge to a trivial solution as the loss function might encourage more of "similarity" within trajectories than "discrepancy" be-

tween trajectories. To address this issue, we propose to use batch normalized latent function during training by replacing $H_{\theta_c}(x)$ with $\overline{H}_{\theta_c}(x)$ with

$$\overline{H}_{\theta_c}(x_j) = \frac{H_{\theta_c}(x) - \mu_{batch}}{\sigma_{batch}} \tag{8}$$

where $\mu_{batch} = \frac{1}{n}\sum_i^n H_{\theta_c}(x_i)$ and $\sigma_{batch} = \sqrt{\frac{1}{n}\sum_i^n(H_{\theta_c}(x_i) - \mu_{batch})^2}$ are the element-wise mean and standard deviation of the mini-batch.

From Figure 4, we use different dimensions (dim= 2 and 1) for the latent space and plot the learned conservation along 6 trajectories. We found that for both dimensions, the learned conservations are distinguishable but struggling to follow the flat line, where the periodic pattern from the orbit loops challenges the prediction power of the neural network to find the correct law. In terms of the relationship between learned and actual conservation, it is difficult to draw any conclusions other than the ordering of the trajectories. Despite not learning the exact conservation law, in the simulation stage, our method still outperforms the vanilla neural network by a large margin in both metrics with the learned invariant. Interestingly, the conservation laws are better enforced with $\dim(H_{\theta_c}(x)) = 2$ comparing to $\dim(H_{\theta_c}(x)) = 1$, indicating the system naturally follows 2 conservation laws.

### 4.3 LARGER SYSTEM: HEAT EQUATION

To further extend our model to larger systems, we test our method on solving the **Heat Equation** on a 1D rod. The 1D rod is given some initial temperature distribution and insulated boundary condition on both ends. The temperature $U(y, t)$, as a function of coordinate and time, gradually evens up following the heat equation $\frac{\partial U}{\partial t} = \frac{\partial^2 U}{\partial y^2}$. The total internal energy along the rod does not vary because the heat flow is blocked by the boundary. We use system state $x$ consisting of overall 101 nodes to discretize $y \in [-5, 5]$ and compress system states to a 9 dimension latent space with an autoencoder pair $(E_{\theta_E}, D_{\theta_D})$. For both contrastive conservation learning and dynamical system learning, original space state and time derivative $(x, \dot{x})$ are mapped to the autoencoder latent space $(z, \dot{z})$ by

$$z = E_{\theta_E}(x) \tag{9}$$

$$\dot{z} = \frac{\partial E_{\theta_E}(x)}{\partial x} \times \dot{x} \tag{10}$$

where $\times$ denotes the matrix multiplication by chain rule, the partial derivative from latent space to original space can be calculated by auto-differentiation package. After simulation, the latent space trajectory will be mapped back to the original space by $D_{\theta_D}$.

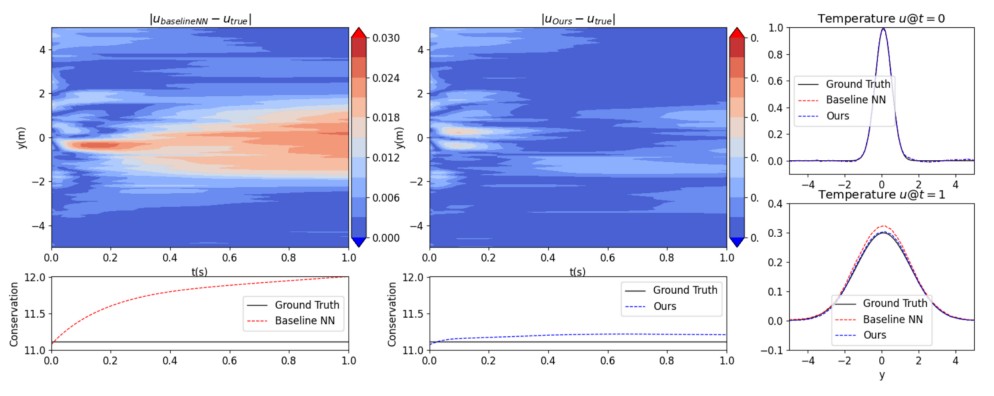

Figure 5: Heat Equation Simulation. Left column: vanilla neural network coordinate error and conservation violation to ground truth. Mid-column: our method. Right column: initial and final temperature distribution

Figure 5 shows the simulation result and conservation metric comparison between vanilla neural network and our method. For both methods, the initial conservation violation error was introduced

by autoencoder. In general, our method conforms to ground truth trajectory and conservation laws much better than the vanilla method.

We generalize the quantitative results for all the experiments above in Table 1. Each of the experiments is performed over three random seeds. During testing, we integrate the trajectory with Runge-Kutta method and compare the trajectory state coordinate and conservation error to the ground truth. Our method outperforms the baseline neural network by a large margin, and the error is often multiple times smaller. One may notice that the standard deviation is comparable with the error metric in the experiments, this is due to the instability of the dynamical system. The trajectory tracking error will exponentially grow as a function of time, and the cases with outlier initialization are likely to dominate the averaged results and lead to large variance. We append the same results in log scale in Appendix A to help clarification. In practice, our method is capable to control the tracking deviation better than the baseline method across almost all the cases.

Table 1: Simulation error over the tasks

| Task | Mean square error | | Violation of conservation laws | |
|---|---|---|---|---|
| | Baseline NN | ConCerNet | Baseline NN | ConCerNet |
| Ideal spring mass system | $0.209 \pm 0.172$ | $\mathbf{0.076 \pm 0.063}$ | $0.096 \pm 0.080$ | $\mathbf{0.002 \pm 0.002}$ |
| Chemical kinematics | $0.064 \pm 0.006$ | $\mathbf{0.031 \pm 0.033}$ | $0.025 \pm 0.012$ | $\mathbf{0.003 \pm 0.001}$ |
| Kepler system | $0.854 \pm 0.103$ | $\mathbf{0.328 \pm 0.026}$ | $0.060 \pm 0.011$ | $\mathbf{0.009 \pm 0.012}$ |
| Heat equation | $0.133 \pm 0.054$ | $\mathbf{0.098 \pm 0.073}$ | $0.686 \pm 0.546$ | $\mathbf{0.178 \pm 0.045}$ |

## 5 DISCUSSIONS

### 5.1 LEARNED INVARIANTS VS EXACT CONSERVATION LAWS

To notice, the learned invariants are different from exact physics conservation laws. From Figure 2, we can tell the learned quantity is approximately a linear function of the conserved quantity. We will give a glimpse of the intuition of why linear function reaches the minimal loss with one-dimensional conservation assumption in Section 5.2. In fact, the linear coefficient is not always positive, therefore the sign of the conservation function is not guaranteed. This is the natural result for a perfectly conserved system, lacking of time evolution information for conservation quantities. In real world cases, many systems are dissipative, with directional time derivative on nominal conservation. We design a ranking loss function to utilize the directional information and encourage the correct sign of the learn invariants, the result is delayed to Appendix B.

Despite the similarity between learned invariants and conservation laws for the above cases, for a general system with more than one conservation terms, the learned invariant can be a non-linear function of the conservation terms. Therefore, it might not preserve the linear relationship. In this paper, we focus on improving the conservation performance for dynamical modeling and leave the task to find the multiple conservation laws to future work. Regardless of the mapping between the learned function and the exact conservation function, the simulation is guaranteed to preserve the conservation property if the mapping is bijection.

### 5.2 CONTRASTIVELY LEARNED LATENT SPACE FOR CONTINUOUS LABELS

As the original contrastive learning framework is designed for discrete labels, we would like to investigate how the learned latent space looks like when a continuous conservation function implicitly labels the observations. Let a system with state $x$ has distribution $D$ on a compact set $\mathcal{X} \subset \mathbb{R}^n$, there exists a continuous scalar function $g : \mathbb{R}^n \to \mathbb{R}$ denoting the unknown conservation function mapping $\mathcal{X}$ to another compact set $\mathcal{Z} = \{z | z = g(x), x \in \mathcal{X}\}$. Consider $\mathcal{Z}$ as a continuous label space, where close examples in the $\mathcal{Z}$ space are considered noisy observations to each other and belong to the same class. Formally, $(x, x') \in \{\text{positive pairs}\}, \forall |g(x') - g(x)| \leq \epsilon$. Then we can write the "continuous" version of the contrastive loss function Equation (3).

$$\mathcal{L}_{\text{continuous}} = - \int_{\mathcal{X}} p_D(x) log\left(\frac{\int_{\{x' | |g(x') - g(x)| \leq \epsilon\}} p_D(x') exp(-(h_\theta(x) - h_\theta(x'))^2) dx'}{\int_{\mathcal{X}} p_D(x'') exp(-(h_\theta(x) - h_\theta(x''))^2) dx''}\right) dx \quad (11)$$

where $h_\theta()$ is the parameterized conservation function and $p_D()$ is the probability over the distribution $D$. The numerator traverses the similarity on the neighborhood $x'$ assumed in the same class of $x$, the denominator calculates the same integral over the entire space. As we attempt to analytically solve the optimization problem for Equation (11) but only come up with a trivial maximum solution, we conduct numerical experiments and make the following conjecture:

**Conjecture 1.** *Let $D$ be a uniform distribution, optimize $\mathcal{L}_{continuous}$ over a bounded set of $\theta$, then $h_\theta(x)$ is an affine function of $g(x)$ (e.g. $h_\theta(x) = c_1 g(x) + c_2$ ) when $\mathcal{L}_{continuous}$ achieves a minimum point.*

We illustrate the conjecture through the following numerical example. We consider a 1D uniform distribution of $x$ over the set $\mathcal{X} = [-1, 1]$ and $g(x) = x$ for simplicity. We parameterize $h_\theta(x)$ with a quadratic function and show the numerical results of $\mathcal{L}_{continuous}$ in Figure 6. The trivial maximum point is when $h_\theta(x)$ being constant (blue). The minimum point reaches when $h_\theta(x)$ is a linear function with the largest absolute coefficient (green). Interestingly, the red solution performs worse than green with the same starting and ending points, indicating a deviation from the linear transform will increase $\mathcal{L}_{continuous}$. For more generality, we provide another numerical experiments with $g(x) = x^2$ without bijection relationship between $\mathcal{X}$ and $\mathcal{Z}$ in Appendix C, the result is consistent with our conjecture.

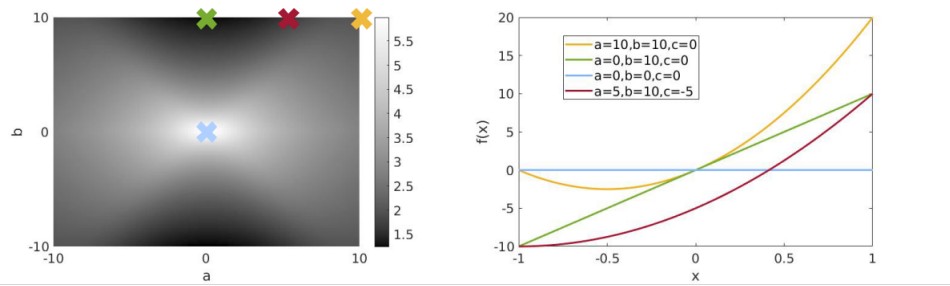

Figure 6: Right: $h_\theta(x) = ax^2 + bx + c$ on different parameterization. Left: Contrastive loss on 1D continuous label space as function of $(a, b) \in [-10, 10]^2, \epsilon = 0.05$.

### 5.3 CONTRASTIVE LEARNING TRAINING LOGISTICS

During the training of trajectory invariants with contrastive learning, we observe certain interesting phenomenon and would like to involve the discussion here for readers' reference. We use the linear regression error to the exact conservation function as metric and delay the experiment results under different hyper-parameters to Appendix D. We found the fitting error decays with $\mathcal{O}(N^{-1/2})$, where $N$ is the training trajectory number. This echoes with the supervised deep learning generalization bound in literature (Yehudai & Shamir, 2019; Cao & Gu, 2019; 2020). In practical contrastive training, we also use small sizes (10-50) as large sizes slightly compromise the performance. Our intuitive explanation is that a large batch is likely to involve similar but less distinguishable trajectories.

## 6 CONCLUSION

In this paper, we propose ConCerNet, a generic framework to learn the dynamical system with designed features to preserve the invariant properties along the simulation trajectory. We firstly learn the conservation manifold in the state space with contrastive view over the trajectory observation, then purposely enforce the dynamical system to stay in the desired subspace by appending a projection layer after the nominal neural network. We show the advantage of our proposed method in both simulation error and conservation metrics and extendibility to be incorporated into larger models. Despite the paper presents an end-to-end approach, both contrastive learning on system invariants and projected dynamical system learning can be seen as an independent procedure and open up a different direction. We believe these ideas represent a promising route in automated system property discovery and practical dynamical system modeling.

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
