# OpenReview forum: "Imposing conservation properties in deep dynamics modeling via contrastive learning"
_ICLR.cc/2023/Conference — Submitted to ICLR 2023_

### Official Review · Reviewer_qJvt · 2022-10-17

**Confidence:** 5
**Correctness:** 2
**Technical Novelty And Significance:** 3
**Empirical Novelty And Significance:** 2
**Recommendation:** 3

**Clarity, Quality, Novelty And Reproducibility:**

- Clarity: See above.
- Quality: The writing requires significantly more effort. Similarly the experiments require justifications and comparisons with baselines. I believe these considerably diminish the quality of the presented work.
- Novelty: The approach brings two rather known objectives together. Yet, the idea of explicitly encoding the conservation as well as the projection (of the dynamics) are timely and would be beneficial for the community.
- Reproducibility: None of the experiment details is visible in the main text.

**Strength And Weaknesses:**

Strengths:
- The model surely learns _some_ constants that remain the same throughout integration.
- The framework is straightforward: Similar contrastive objectives are commonly used and well understood. The dynamics function is also easy to understand as it is built to preserve the constant variables.

Weaknesses:
- The paper suffers from clarity issues as well as grammatical mistakes. I believe it should undergo significant updates. See my comments at the end of this question.
- Methodology should be motivated. The authors argue that "this is the first work that studies discovery of conservation laws for general dynamical systems through contrastive learning." What if we had a simpler method in which certain global parameters remain constant throughout integration? Would the proposed approach still be preferable? If so, why? Furthermore, among all other techniques, why is contrastive learning chosen? It does come with a complicated and costly loss term; so what is the catch here instead of a simpler approach?
 - The gradient matching term (eq. 7) should be carefully analyzed. It is well known that this objective becomes problematic with increasing noise levels and bigger time differences as the differential function is directly fitted to the empirical gradients $\frac{x_{i+1}-x_{i}}{t_{i+1}-t_{i}}$. This is in contrast to learning a generative process, such as GP or neural ODEs, and is the very motivation of these approaches.
- The paper lacks a whole discussion on causality. In my understanding, the method discovers certain factors on which the trajectory (forward simulation) is conditioned. This is the very definition of causal learning while the paper does not touch upon this connection. Similarly, the connections between invariance and causality could be included (in addition to the short discussion on invariance that is already included).
- The method should be compared against SOTA baselines. I do acknowledge that the literature on "black-box conservation learning" is quite limited but at least Lagrangian NNs or some adaptation of vanilla neural ODE systems could be considered here. This would help us judge the level of difficulty of the problem. Similarly, we should see how the performance would change if we consider higher dimensional latent spaces ($m>2$) in Sec 4.2.


Writing issues:
 - "deploying the knowledge towards the new occurrence" is unclear.
 - "conservation law were" is wrong
 - The connection between the sentence starting as "On the other hand, data-driven" and the prior sentences is missing.
 - "one or more distinguishing features of physics-based system trajectories" This sounds mysterious, i.e., what are the examples of such features?
 - "By comparing the latent space distance of the system state observation and assigning them into their trajectory" Both parts of the sentence is unclear. What is latent space distance? How does "assignment" take place?
 - What does "ConCerNet" stand for?
 - "of the simulation in the longer term" Longer compared to what?
 - Naming the function approximations already on the first page would help follow the text.
 - "projecting the dynamical neural network output towards the" Perhaps "on the"?
 - "conservation manifold from the first one" is unclear. Perhaps something like "the manifold learned by the first module"?
 - "guaranteeing" instead of "guarantying"
 - "Unlike discriminative models that explicitly learn the data mappings" What mapping is referred to here?
 - Sec2.1 could be organized chronologically (by moving the paragraph starting with "Despite the recent surge of" to the beginning)
 - Neural ODE paper (Chen, Ricky TQ, et al. "Neural ordinary differential equations." Advances in neural information processing systems 31 (2018).) should be cited in Sec 2.2.
 - "however, the soft Lagrangian treatment does not guarantee the model performance during testing" This should be elaborated.
 - "to propose a contrastive learning framework in a more generic form that is compatible with arbitrary conservation." What is meant by "arbitrary conservation"?
 - "total time step T" is not clear.
 - "We let the trajectory set ... be the simulation history starting". I'm not sure what this means: Do $C_k$'s refer to sequence ids?
 - "the initial conditions have various conservation values." What does this exactly mean?
 - "where the probability of point ... **belonging** to the trajectory"
 - Similar to a previously raised concern: What is the motivation for eq.2? What does it imply?
 - \log and \exp should be used instead of log and exp.
 - What does $x\sim C \sim D$ mean?
 - A short discussion on NCA would be nice as it is referred to multiple times.
 - "Softmax like function, ensuring the nice property of probabilistic distribution" What does this mean? What is nice here?
 - "learning of the m conservation terms" Perhaps an m-dim vector is referred to?
 - "by eliminating its parallel component" Parallel to what?
 - Eq,1 uses $f_\theta(x)$ whereas it later becomes $\tilde{f}_{\theta_d}(x)$. What is the difference?
 - Eq. 5-6 are unclear. A (verbal), more intuitive explanation should be given. Projection(.) function should be defined.
 - "the orthonormalized set of vectors from" is unclear.
 - "conservation term in learned"
 - x and y axes names and figure titles should be added/revisited.
 - What are the system states in Sec 4.1?
 - Why to use $x[1]$ instead of $x_1$?
 - "the learned conservation here is approximately the exact conservation" is unclear.
 - "coordinate mean square loss to the ground truth" is unclear.
 - Are the equations in the first paragraph of 4.2 really difficult to learn, given the recent success of neural nets on many tasks?
 - Why does the issue that is resolved by eq 8 take place in the first place? Why only on this experiment? Why does eq.8 help at all?
 - Better to use $m=2$ instead of $dim(H_{...})=2$.
 - Table1 could be given at the very beginning of Sec4 as it nicely summarizes all results.

**Summary Of The Paper:**

This paper presents a technique to preserve physical conservation laws in neural network based dynamical systems. The authors propose to solve two tasks concurrently: (i) learning variables that remain constant for a given trajectory, and (ii) a dynamics function that respects the constant variables and that fits input trajectories. The first objective is achieved by adopting well-known contrastive learning objectives so that the new objective "pulls the latent factors" (constant) within a trajectory whereas it "pushes them away" if they belong to different sequences. The second goal is realized by a gradient-matching loss. The model is experimentally demonstrated to preserve constant terms oftentimes and hence leads to non-divergent trajectories.

**Summary Of The Review:**

I believe encoding state-invariant variables is a nice idea that comes with immediate real-world applications. A contrastive objective to learn such variables is shown to be effective. However, we are unable to tell whether the benchmarks are sufficiently challenging (so that we can see the limits of the framework). We are also not in a position to decide whether simpler ideas or existing methods would perform any better. Moreover, the text requires significant updates, which is why I recommend a reject.

---

> ### Author Response · Authors · 2022-11-14
> **author rebuttal (1/4)**
>
> Dear reviewer qJvt:
>
> Thank you for the feedback and pointing out your concerns, they are of great value to us, and we are very glad to clarify them in the following. As a short summary of the detailed paragraph below, we would like to clarify, 1.  We have modified the paper extensively following your suggestions, and all the points you mentioned are replied one by one below. 2. Following your suggestion, we added HNN as comparison baselines in two of our experiments, as it is not applicable to two other non-Hamiltonian systems. We also add parametric study on latent space on the heat equation. 3. As we are confused by your question on the methodology motivation, please explicitly point out what simpler method can we use to achieve the same goal.
>
> We hope our answers address all your concerns. Please do not hesitate to contact us if you need any additional clarifications. We will do our best to resolve all your concerns and wish you will re-evaluate our paper.
>
>
> Weakness:
>
> **$\bullet$ 1. Grammar issue**: thank you for pointing it out, we made significant changes and reply to your questions one by one in the following.
>
> **$\bullet$ 2.  Methodology motivation:** Honestly, we are confused by your questions **“What if we had a simpler method in which certain global parameters remain constant throughout integration?”, “so what is the catch here instead of a simpler approach?”  as you didn’t explicitly point out what “simpler” approach we can use**. Please provide such methods if possible. For the first question, our projection idea is already a simple way to do that, neural ODE will not help. For the second question, if you mean there exist other methods to discover conservation laws, the prior work all require heavy pre-processing and artificial knowledge on fitting function class to acquire the conservation laws (i.e. Ref [5] [6] requires physics information such as symmetry and a brutal force search over combinations of function class). However, our contrastive learning method does not require any prior information and uses a general neural network as parameterization instead of given function classes. This is the major motivation of our approach.
>
> **$\bullet$ 3. gradient matching term:** This could be problematic in real-world scenario if the observation time step is too large and neural ODE could help as it serves as a differentiable integrator in training. However, this is an independent and parallel research field to our work. Directly using system time derivation is a common way in related works such as HNN and LNN (Ref 4,7). In our implementation, all the data is generated with random noise which counts in the factor of empirical gradients. Besides, either our method or previous work like HNN or LNN can be coupled with Neural ODE or other discretization methods to get more accurate estimation of the data.
>
> **$\bullet$ 4. Lacking discussion on causality:** causal learning is a less related research field to our work. In the dynamical system we modeled, the system evolution only depends on the current system state, rather than causality from previous steps. In the prior works encouraging invariants in dynamical system modeling (Ref 4,7), there is no discussion on causal learning as well.
>
> **$\bullet$ 5. comparison against SOTA baselines:** Thank you for pointing it out. During the rebuttal time, we have done comparison experiments with HNN (Ref 4) for the spring mass system and Kepler examples in Appendix A Table 4. The LNN is still an arxiv paper before peer review and the implement is based on JAX. Therefore, we do not include it as comparison baseline. For the other 2 examples, both HNN or LNN are not applicable as they are not Hamiltonian systems. Regarding neural ODE, as we explain above, this is a parallel research field and does not help in preserving quantities in dynamics modeling.
>
> Refs:
>
> [1] Debiased Contrastive Learning. Ching-Yao Chuang, Joshua Robinson, Lin Yen-Chen Antonio Torralba, Stefanie Jegelka. NeurIPS 2020
>
> [2] Ching-Yun Ko, Jeet Mohapatra, Sijia Liu, Pin-Yu Chen, Luca Daniel, and Lily Weng. Revisiting
> contrastive learning through the lens of neighborhood component analysis: an integrated framework. ICML 2022
>
> [3] J Zico Kolter and Gaurav Manek. Learning stable deep dynamics models. Advances in neural
> information processing systems, 32, 2019
>
> [4] Hamiltonian Neural Networks. Samuel Greydanus, Misko Dzamba, Jason Yosinski NeurIPS 2019
>
> [5] Liu, Ziming, and Max Tegmark. "Machine learning conservation laws from trajectories." Physical Review Letters 126.18 (2021): 180604.
>
>
> [6] AI Feynman: A physics-inspired method for symbolic regression. SM Udrescu, M Tegmark - Science Advances, 2020
>
>
> [7] Miles Cranmer, Sam Greydanus, Stephan Hoyer, Peter Battaglia, David Spergel, and Shirley Ho. Lagrangian neural networks. arXiv preprint arXiv:2003.04630, 2020

---

> > ### Author Response · Authors · 2022-11-14
> > **author rebuttal (4/4)**
> >
> > $\bullet$ "the learned conservation here is approximately the exact conservation differing by some constant coefficient" is unclear.: by this sentence, we mean the learned function is an affine function of the exact physics conservation. A similar description was in previous papers (Ref [4]):  “one can see that the HNN-conserved quantity has the same scale as total energy, but differs by a constant factor”.
> >
> > $\bullet$ "coordinate mean square loss to the ground truth" is unclear. We mean the l2 distance between the prediction and ground truth. We rewrite it to “coordinate error” for clarification.
> >
> > $\bullet$ equations in the first paragraph of 4.2: The square root on the denominator is difficult to be represented by the simple neural network structure (1 hidden layer) we use in the paper. Besides, there exist numerical issues due to high conditional numbers (the expression is very sensitive to x[3], x[4] to x[1], x[2]) for training as we don’t apply any normalization. We tested a supervised method to learn the function and the results are not better. Besides, in our proposed methods, we can only learn a function (i.e. linear combination) of the two conservation functions, instead of the exact function. We include the discussion in Sec 5.1.
> >
> >
> > $\bullet$ Eqn 8: The issue we met here is that the optimization falls into some trivial solution,  such that all the $H_\theta(x)$ outputs a similar value no matter what input $x$ is. We guess this happens in this experiment because the model does not have the capacity or enough data to learn a complex conservation formula even for the points on the same trajectory, and the optimization can fall into some local minimum. In Eqn (3), both minimizing the positive pair difference and enlarging the negative pair difference will decrease the loss function. If the model cannot find a constant output for all the points in the same trajectory (this Kepler example, Fig 3 upper row), the model will tend to emphasize encouraging the similarity between positive pairs first, but neglects to or cannot handle the dissimilarity between negative pairs. To address that, we use the batch normalized distance instead of the absolute distance. If the model learns some trivial or similar values, the optimization can jump out of this local minimum because the relative distance instead of the absolute distance is used in the loss function.
> >
> > $\bullet$ Better to use m=2: Thank you for the suggestions, we use $dim(H)=2$ in the plots to facilitate readers, as they may forget the definition of $m$.
> >
> > $\bullet$ Table1 could be given at the very beginning of Sec4: Table 1 is the summary of the results of the experiments. In Sec 4 we need to present the details of the dynamics, experiments setup, warm-up examples, intermediate results, and evaluation metrics before the final results. It would be confusing to abruptly give a result summary without any introduction.
> >
> > $\bullet$ Reproducibility: None of the experiment details is visible in the main text.: We listed all the experiment details in Appendix A and we set a reminder for readers to reroute in sec 4.

---

> > ### Author Response · Authors · 2022-11-14
> > **author rebuttal (3/4)**
> >
> > $\bullet$ "where the probability of point ... belonging to the trajectory": thank you for pointing out the typo, we have corrected it.
> >
> > $\bullet$ motivation of eq.2: Eqn 2 provides a probabilistic view of our constrastive training method, where we aim to classify the point into different trajectories. This probabilistic view is common in contrastive learning papers (Ref [1,2]). Eqn 2 is critical for readers to understand the find loss function design (Eqn 3).
> >
> > $\bullet$ “\log and \exp”: thank you for pointing it out, we have corrected it.
> >
> > $\bullet$ “$x\sim C\sim D$” Thank you for pointing out the confusion, by $x\sim C\sim D$ we meant the anchor point $x$ is sampled from a trajectory $C$ and each trajectory $C$ is drawn from a given distribution $D$. To better clarify, we rewrite the formulation as $x\sim C, C\sim D; x^+\in C_{\backslash x}; x^- \in C^-, C^- \sim D_{\backslash C}$. Other than the anchor point $x$, its positive pairs ($x^+$) are drawn from the relative complement set of $x$ in $C$. The negative pairs ($x^-$)  are drawn from other trajectories.
> >
> > $\bullet$ Discussion on NCA: we cite NCA as an inspiring example but do not include a discussion on its details because our method is more related to modern contrastive learning like Ref [2], instead of NCA. For the description of NCA method, readers can refer to the original NCA paper we cited. For the connections between NCA and contrastive learning, Ref [2] provides a detailed discussion.
> >
> > $\bullet$ "Softmax like function, ensuring the nice property of probabilistic distribution": By using the softmax like function, the outputs automatically sum to 1. This is good for probabilistic interpretation.
> >
> > $\bullet$ "learning of the m conservation terms" : it means the learned m-dim vector **$H_{\theta_c}(x)\in\mathbb{R}^m$** which is in this sentence, right after "learning of the m conservation terms". The definition of $H_{\theta_c}(x)$ is in the 3rd line under sec 3.1.
> >
> > $\bullet$ "by eliminating its parallel component" Parallel to what?:  We mean the component is parallel **to the normal direction of the invariant planes (i.e. $\nabla_x H_{\theta_c}(x)$)**. The bold description is this sentence, right after "by eliminating its parallel component".
> >
> > $\bullet$ $f_\theta(x)$ to $\tilde{f}_{\theta_d}(x)$:
> >
> > $f_\theta(x)$ in Eqn 1 (sec 2: background and related work) is the vanilla neural network in the problem formulation for general dynamical system with parameter $\theta$. $\tilde{f}_{\theta_d}(x)$ is our proposed dynamical system prediction after the projection layer (Eqn 5,6, sec 3 Proposed methods). The tilde sign indicates projection, and we use another parameter notation $\theta_d$ to differentiate with other neural network parameters used in this method.
> >
> > $\bullet$ Eqn 5,6 are unclear: The projection function is defined in Eqn 5 (the first term denotes the vector to be projected, and the second term is projective space). This convention is common in previous related papers (Ref [3]). The verbal description is one sentence above. **we can project the nominal neural network output $f_{\theta_d}(x)$ onto the conservation manifold by eliminating its parallel component to the normal direction of the invariant planes (i.e. $\nabla_x H_{\theta_c}(x)$)**
> >
> > $\bullet$ "the orthonormalized set of vectors from" is unclear: it means the set of orthonormalized vectors that are generated from a given set of vectors
> >
> > $\bullet$ "conservation term in learned": thank you for pointing out the typo, we corrected it.
> >
> > $\bullet$ “x and y axes names”: thank you for pointing this out, we have modified it.
> >
> > $\bullet$ system states in Sec 4.1: the system state is denoted as $x$ throughout the paper. For the description of each experiment, we provided a list in Appendix A. We also set a reminder for readers to reroute to Appendix A in sec 4 (1 sentence above the title of sec 4.1).
> >
> > $\bullet$ why use $x[1]$ instead of $x_1$?: Because the subscript in our paper indicates time index and superscript means trajectory index. Using $x_1$ will confuse the readers. By  $x[1]$ we mean the first element in the vector  $x$, this convention is easy to understand for the ML community.

---

> > ### Author Response · Authors · 2022-11-14
> > **author rebuttal (2/4)**
> >
> > **$\bullet$ Writing issues** : thank you for pointing out our writing issues, we agree with most of them and mark the modification in red in the new manuscript. Below is our reply to each point.
> >
> > $\bullet$ "deploying the knowledge towards the new occurrence" means ML models apply the learned knowledge to the unseen data, this is clear here.
> >
> > $\bullet$ "conservation law were": we corrected it.
> >
> > $\bullet$ The connection between the sentence starting as "On the other hand, data-driven" and the prior sentences is missing.: The prior sentence to "On the other hand, data-driven dynamical modeling is prone to violation of physics laws or instability issues" is “However, such discovery usually requires extensive human insights and customized strategies for specific problems.” These are the two disadvantages when data-driven method is used for physics discovery.
> >
> > $\bullet$ "one or more distinguishing features of physics-based system trajectories": we explain this in the next sentence: “By comparing the latent space distance of the system state observation and assigning them into their trajectory, we aim to learn a $low-dimensional representation$ potentially serving as the invariant term for the system”. Following your suggestion, we added the example description “i.e. conservation values” in the first sentence.
> >
> > $\bullet$ "By comparing the latent space distance of the system state
> > observation and assigning them into their trajectory, we aim to learn a low-dimensional representation potentially serving as the invariant term for the system": “latent space” means the low-dimensional representation space from the second half of the sentence, “distance” is a common math definition to compare two points in a metric space. We agree with your confusion in “assigning them into their trajectory” and we delete it.
> >
> > $\bullet$ "ConCerNet" is short for CONtrastive ConsERved Network, as we include the full spelling in the 2nd point of our contribution summary (page 2).
> >
> > $\bullet$ "of the simulation in the longer term": we rewrite it as "of the simulation in the long term prediction"
> >
> > $\bullet$ “Naming the function approximations already on the first page would help follow the text.”: There is no function approximation in the first page. If you mean $f(x)$ in Figure 1 page 2, we explain it in Eqn (1) in page 3. Because page 1,2 are mainly for introduction, Figure 1 is an illustrative diagram to draw the big picture of our method pipeline, we believe it is appropriate to explain the functions in page 3.
> >
> > $\bullet$ "projecting the dynamical neural network output towards the": we correct it to “on the"
> >
> > $\bullet$ "conservation manifold from the first one": we correct it to "conservation manifold learned by the first module"
> >
> > $\bullet$ "guarantying": we correct it to "guaranteeing"
> >
> > $\bullet$ “Unlike discriminative models that explicitly learn the data mappings”: discriminative models refer to a class of models mainly for supervised learning like regression and classification, here “data mappings” means mapping between data to label.
> >
> > $\bullet$ Sec2.1 organization: Thank you for the suggestion, we reorganized this section chronologically.
> >
> > $\bullet$ citing Neural ODE paper. In sec 2.2, we introduced NN-based dynamical system modeling problem formulation in the first paragraph and recent works to encourage the conservation laws during prediction in the second paragraph. Neural ODE can be seen as a differentiable integrator, this is an independent parallel research field, i.e. Neural ODE can be used with the mentioned works in this section, but Neural ODE is not alternative to the cited works, nor does it change the modeling problem. Therefore, we think it is inappropriate to cite Neural ODE paper here.
> >
> > $\bullet$ "however, the soft Lagrangian treatment does not guarantee the model performance during testing": adding another loss term is the Lagrangian multiplier method to solve constrained optimization, it does not guarantee the constraint in the testing data.
> >
> > $\bullet$ "to propose a contrastive learning framework in a more generic form that is compatible with arbitrary conservation.": "arbitrary conservation" means any conservation without underlying structure limit, such as Hamiltonian. We also do not assume any prior physics knowledge for the conservation.
> >
> > $\bullet$ "total time step T" is not clear: it means the number of total time steps in one trajectory is T.
> >
> > $\bullet$ $C_i$, yes $i$ is the trajectory/sequence id, denoted in the 2nd line of the 1st paragraph under section 3.1.
> >
> > $\bullet$ "the initial conditions have various conservation values.": it means when we draw the initial conditions to produce the trajectory, the conservation values of initial conditions are different. For example, I can draw a few random ideal spring mass system states, with different total mechanical energies.

---

> > ### Comment · Area_Chair_v8iL · 2022-11-22
> > **Follow up on rebuttal**
> >
> > Dear reviewer,
> >
> > the authors tried to address the points you raised in your review, and there seems to be an unclarity regarding motivation. Can you check if the rest of the response is ok, and clarify how the authors should interpret your question regarding a simpler method with constant global parameters?
> >
> > Cheers,
> > Your AC

---

> > ### Comment · Reviewer_qJvt · 2022-11-24
> > **response to rebuttal**
> >
> > [Dear AC, do you know if the authors will be able to see this message?]
> >
> > Hello again,
> >
> > Thanks for your detailed rebuttal. I'm sorry for the confusion and let me express my concern clearly. The presented approach is tested on a series of benchmarks with increasing complexity. The conservation laws in the first three benchmarks (spring mass, chemical kinematics, and Kepler) are so straightforward that I believe even much simpler approaches would be sufficient. Two examples: For the Kepler system, one could simply learn a variable that stays constant throughout integration, which would be a very simple implementation of "angular momentum conservation". Alternatively, for the chemical reaction, one can augment the loss with a simple regularization term, e.g., $\sum_i \sum_j || x(t_i)-x(t_j) ||_2^2$.
> >
> > That being said, the approach describes a generic recipe and thereby prevents us from designing problem-specific "conservation objectives". This is one of the aspects of the work that I now believe I have underestimated. Yet, I believe only the last application is worth dealing with this contrastive learning objective, which can come up with additional hassle and computational cost, instead of simple regularizations. I believe a more convincing example case would be pendulum/Kepler system with angular position/velocities, which implies a much more complex energy conservation law. Similarly, it would be very interesting to have a complex system with momentum conservation, e.g., an elastic collision of bodies with different masses.

---

> > > ### Author Response · Authors · 2022-11-24
> > > **Response to reviewer qJvt**
> > >
> > > Dear reviewer:
> > >
> > > Thank you for the timely response and clarifying your concerns.
> > >
> > > We agree with you that the conservation laws for the first three examples are indeed simple in terms of their equation formats, however, we believe it is still difficult to learn or enforce in the dynamics simulation without any physics prior knowledge. For the chemical reaction, we guess your regularization terms penalizes the mass difference between different states within one trajectory. While, there are two issues here: 1. The dynamics observation data is the only information during training, we do not know such problem-specific regularization functions (i.e. summation of mass).  2. By augmenting the regularization loss, we solve the optimization problem with a "soft" constraint, which might improve the statistical performance of conservation in predicted trajectory.  While this approach has nothing to say in terms of guaranteeing the property. Differently, our projection method is a structural "hard"constraint that enforces the conservation of the learned function in prediction.
> > >
> > > Regarding the Kepler example, we are still a bit confused on how to **simply learn a variable that stays constant throughout integration** ? Naive approach can easily learn a constant function regardless of input. Besides that, we would like to re-clarify that it is difficult to learn the exact angular momentum function for the Kepler example without prior knowledge. Because there exist more than 1 conservation laws in the system, any combinations of the exact conservation functions are conserved along the trajectory. A simple example is the summation of the energy function and angular momentum function, it is conserved as well.
> > >
> > > We are glad to hear that you recognize our generic approach to the problem. We also appreciate your suggestion on the collision of bodies example, and we believe it could be an excellent illustrative example (maybe we need to modify it to remove the energy conservation in order to find the momentum conservation). As we are not allowed to modify the manuscript anymore in the review session, we are aiming to conduct the experiments by this weekend and post the results here in the openview reply window. We will add the experiments to the final version of the manuscript.
> > >
> > > Thanks,
> > >
> > > Authors

---

> > > > ### Comment · Reviewer_qJvt · 2022-11-24
> > > > **.**
> > > >
> > > > I think the discussion is really going out of hand, I hope this finally clears the dust: For a closed system with two particles with equal masses, one may define the $x$-velocities of the particles as $v_x(t)$ and $C-v_x(t)$, which would automatically ensure momentum conservation. Here, $C$ is a very simple, sequence-specific constant. If I understand correctly, this (or a similar) construction would also apply to ~spring mass~ chemical reaction system.
> > > >
> > > > My "a variable that stays constant throughout integration" suggestion is by no means my main point, which I am obviously unable to express clearly. So again; all I'm trying to say is simple state constraints such as the ones that I recommended or similar others could be sufficient to achieve the conservations you considered. These should have been investigated in the paper. When compared only against a standard NN, I don't see whether your method is actually a breakthrough or a quadratic regularization term could also achieve the same.
> > > >
> > > > I disagree with your vague "soft" and "hard" constraint taxonomy. After all, your "hard" formulation relies on a contrastive loss + projection, where the former is clearly not guaranteed to work off-the-shelf and it may require certain tricks as in eq.8. Consequently, I don't see any "hard" constraint such as the ones in HNNs or second-order systems, which are directly imposed on the architecture or differential function.
> > > >
> > > > I'm not sure if authors are allowed to share additional results after the rebuttal period. If allowed, I would suggest you perform experiments with varying noise levels and observation spacings, e.g., irregularly observed sequences and sequences with small/medium/large observation time differences. On a related note, I'm not able to see the noise levels considered in the experiments. Can you point me to the correct page?

---

> > > > > ### Author Response · Authors · 2022-11-25
> > > > > **Response to reviewer qJvt**
> > > > >
> > > > > Dear reviewer,
> > > > >
> > > > > Thank you for the quick response, and we apologize for the long response, as we are trying to address all your concerns and hope to clarify some misunderstanding. Please allow us to address your points in the following:
> > > > >
> > > > > 1. As you mentioned, $C$ is a sequence-specific (actually state-dependent) and we can't use a same $C$ for all the trajectories. Then how to calculate $C$ from either states or sequence is not trivial, our conservation learning is trying to find out such mapping. Now suppose we formulate $C$ as $C_\theta(x)$ and use it in the way you suggested (define velocity as $v$ and $C-v$), this formulation only works for the chemical reaction example of the 4 dynamical system we provided. Such prior knowledge or case dependent formulation can be hardly generalized into other examples.
> > > > >
> > > > > 2. The **simple state constraints** you suggested indeed reflects certain prior knowledge of the problem, therefore it is not fair to compare with our generic approach. Suppose $x_1$ and $x_2$ are the two system state,  $x_1+x_2= C$ looks simple, but the question is why do we choose summation instead of other numerous combinations between $x_1$ and $x_2$ (i.e. $x_1-x_2= C, x_1/x_2= C, x_1*x_2= C...$). If we want to make it compatible to many systems, then it requires either brute force search or a large span of possible combinations and could easily cause over-fitting. Our contribution lies on using a NN as basis function for these combinations and contrastive learning to find out the good representation. The simple constraints or quadratic regularization term would not work for general dynamical system.
> > > > >
> > > > > 3. We insist on our claim of the "hard" and "soft" constraint. For both HNN and our method, there are two learning objects. The first learns a conservation quantity ($x\rightarrow C$), the second models the dynamics. In the first phase, suppose HNN learns the Hamiltonian, then through their dynamics structure $C$ is guaranteed constant along the prediction trajectory. This is the same for our case. If $C$ is well-learned or the exact conservation function is given, despite training or testing phase, the integrated ODE trajectory by $\frac{dx}{dt}=\tilde{f}_\theta(x)=f_\theta(x)- \langle f_\theta(x),\frac{\nabla_x C(x)}{||\nabla_x C(x)||}\rangle \frac{\nabla_x C(x)}{||\nabla_x C(x)||}$ forces $C(x)\equiv \text{constant}$ as the deviation components is eliminated. We do have the guarantee **directly imposed on the architecture or differential function**. The quadratic regularization can never ensure that even we know the exact conservation function. This is the **major structural difference** between "hard" and "soft" constraints. As of your concern to the learning function performance, both HNN and our method are statistical estimation of the exact function on the data distribution. There are differences in performance, but none of them is the exact function. Eq.8 is more of a standard treatment for complex functions instead of a case-specific trick, it can be applied to other examples as well, but they are not replying to it.
> > > > >
> > > > > 4. We think it is ok to share results as long as we don't modify the manuscript. @AC, can you confirm on this?
> > > > > In the current manuscript, a uniform noise is used, and we do not vary its level. Many dynamics learning paper (i.e. HNN) do not involve noise level and observation spacing parametric study, we are happy to conduct the experiments and trying to provide it. But these affiliate studies along with irregularly sequences are parallel research fields and common for general dynamics modeling problems, as our contrastive learning part does not depend on the temporal relationship.
> > > > >
> > > > > Thanks,
> > > > >
> > > > > Authors

---

> > > > > > ### Comment · Reviewer_qJvt · 2022-11-25
> > > > > > **.**
> > > > > >
> > > > > > I think we are again getting lost in details. Just to clarify, I would not mind if a regularization scheme that involves problem-specific knowledge outperforms the presented approach (although that would be surprising to me). Such an experiment would only aim at illustrating the level of difficulty of the problems being solved. It could be very well some other technique to impose conservations, not necessarily regularization. The funny neural ODE baseline shows the lower bound on the acceptable level of performance, what is the upper bound? How would simple and/or problem-specific schemes perform? People often time perform ablations in which they replace certain unknowns with true values just to demonstrate the upper bound, right? Since this paper introduces not just a new model but also a new perspective, it is important to see its limits.
> > > > > >
> > > > > > Again, I'm feeling that we are getting lost in the "hard or soft discussion". Let me clarify my take without using these words as we both use the terms vaguely. As you highlighted, the presented approach would conserve energy if **$C$ is well-learned**. That is, whether the correct quantity is conserved or not relies on successful contrastive learning (CL) training. Please note that I do acknowledge that the presented approach would conserve _something_  regardless of the CL part - eq5&6 already say that. Nonetheless, regardless of how training went, HNN would guarantee energy conservation because this is built-in. I believe this is possible only because HNN preserves a single quantity, unlike your more generic approach. In that sense, I see your approach as a nice attempt between standard ODEs and HNNs (or similar), where learning conserved quantities is a goal, unlike these two extremes.
> > > > > >
> > > > > > Concerning eq.8, is it used in all the experiments? I thought you didn't but now I understand that it is a more generic treatment, then it would make more sense to me to use it in all problems.
> > > > > >
> > > > > > Let me ask again more clearly: What is the standard deviation of noise? Is it the same in all experiments or does it change depending on the magnitudes of state observations?
> > > > > >
> > > > > > Finally, two comments concerning your point: _... these affiliate studies along with irregularly sequences are parallel research fields and common for general dynamics modeling problems, as our contrastive learning part does not depend on the temporal relationship_. I'm glad that you acknowledge a limitation of your method in the last part of your sentence. Also, the very generic title of the paper ("... in deep dynamics modeling ...") implies that this approach is also applicable to irregular sequences. Please correct me if I have misunderstood.

---

> > > > > > > ### Author Response · Authors · 2022-11-25
> > > > > > > **response to reviewer qJvt (3)**
> > > > > > >
> > > > > > > Dear reviewer:
> > > > > > >
> > > > > > > Sorry for the lengthy reply, and thank you for being straight-forward in the response. I will try to be straight-forward as well.
> > > > > > >
> > > > > > > 1. If you are talking about ablation study then that's a whole different story. Our previous claim is trying to convince you the simpler methods you implied in the very first comment do not exist under our setting (at least from our knowledge). The vanilla NN we used in the paper is the lower bound. Neural ODE is not the lower bound, but I don't want to go to details here. Involving prior knowledge, an obvious upper bound is we know the exact dynamics equation $\frac{dx}{dt}$, then the training is meaningless. A fair upper bound is coupling the known conservation function with our projection method. We will try to provide such experiments.
> > > > > > >
> > > > > > > 2. Your understanding on the HNN is wrong. Same as our method, HNN only guarantees the property it learns, but not the exact energy function. Its performance to preserve energy also depends on how good the learned function approximates the energy function. To correct your claim on HNN performance **regardless of how training went**, we provide an extreme counter example: can HNN still preserve the spring mass energy if it never sees the data?
> > > > > > >
> > > > > > > 3. Eqn 8 is not used on all examples, but required on Kepler example to avoid optimization stuck in local minimum. We decide to keep it simple for other cases.
> > > > > > >
> > > > > > > 4. We are using Gaussian noise across all observations with standard deviation = 0.01 in training data and 0 noise on testing data. It is independent of experiments and magnitude.
> > > > > > >
> > > > > > > 5. Yes, we agree the corresponding points are applicable to dynamics modeling and so our paper. And there is a textbook of other dynamical system modeling issues (i.e. uncertainty, observability, discretization scheme) we haven't discussed in the paper. To keep things simple, recent deep dynamics learning papers to preserve quantities (Ref [1][2][3]) along with our paper all follow the simplest setting. As a fair analogy, we are engine suppliers to improve car fuel economy. We follow the industry guideline and compete our engines with other suppliers' on the same car with the same tire. We know the tire choice will surely impact the fuel consumption and the trend will be same for all engines, do you think we need to try all different tires to prove our work?
> > > > > > >
> > > > > > > Thanks,
> > > > > > >
> > > > > > > Authors
> > > > > > >
> > > > > > > [A]: Learning stable deep dynamics models. J Zico Kolter and Gaurav Manek. NeurIPS 2019
> > > > > > >
> > > > > > > [B] Hamiltonian Neural Networks. Samuel Greydanus, Misko Dzamba, Jason Yosinski NeurIPS 2019
> > > > > > >
> > > > > > > [C] Lagrangian Neural Networks. Miles Cranmer, Sam Greydanus, Stephan Hoyer, Peter Battaglia, David Spergel, Shirley Ho. 2020

---

> > ### Author Response · Authors · 2022-11-28
> > **Follow up experiments to reviewer qJvt (1/2)**
> >
> > Dear reviewer qJvt:
> > During the weekend, we have conducted the following additional experiments and would like to share with you. We reported the average results over 5 random seeds across all the experiments.
> >
> > 1. Ablation experiment with known conservation function for the ideal spring mass system. We assume the conservation function $H(x)=x_1^2+x_2^2$ is known, and provided 2 ways to learn the dynamics: 1. Use the projection method we proposed in the paper. 2. Appending an additional loss term $\lambda ||f_\theta(x)^\top \frac{dH}{dx}||^2$ where $\lambda$ is the coefficient and $f_\theta(x)$ is the neural network to approximate the dynamics $\frac{dx}{dt}$. For the regularization method, we provide 3 $\lambda$s.
> >
> > \begin{array}{c| c|c}
> > \hline
> > \hline
> > Method & MSE & \text{Vio. of Conservation} \\
> > \newline
> > \hline
> > \text{ConCerNet (Projection, learned conservation)} & 0.076 & 0.002 \\
> > \newline
> > \hline
> > \text{Vanilla NN} & 0.209 & 0.096 \\
> > \newline
> > \hline
> > \text{Projection, given conservation} & 0.011 & 8.11101e-12 \\
> > \newline
> > \hline
> > \text{Vanilla NN, $\lambda=0.01$, given conservation} & 0.204 & 0.092\\
> > \newline
> > \hline
> > \text{Vanilla NN, $\lambda=0.1$, given conservation} & 0.315 & 0.110 \\
> > \newline
> > \hline
> > \text{Vanilla NN, $\lambda=1$, given conservation} & 0.269 & 0.074\\
> > \newline
> > \hline
> > \end{array}
> >
> > As you can see from the table 1, if we are given a conservation function, the projection method we proposed can guarantee the conservation property (to the accuracy of **$8e-12$**) along the simulation trajectory. This is a simple and effective way to force the “hard” constraint. Using the “soft” regularization is not effective in reducing the conservation violations, despite we tried different regularizer coefficient.
> >
> > We also want to clarify that the projection experiments **with given conservation function** reflects the upper bound performance of our proposed method, but not the difficulty of the original problem. However, we think this method could be useful for real-world complex problem if some rule of thumb conservation function is known like mass conservation for fluid modeling.
> >
> > 2. We conducted the ideal spring mass system experiment under different noise settings with Gaussian noise of standard deviation $\sigma=0.01,0.05, 0.1, 1$.
> >
> > \begin{array}{c| c|c}
> > \hline
> > \hline
> > Method & MSE & \text{Vio. of Conservation} \\
> > \newline
> > \hline
> > \text{ConCerNet, $\sigma=0.01$} & 0.076 & 0.002 \\
> > \newline
> > \hline
> > \text{Vanilla NN, $\sigma=0.01$} & 0.209 & 0.096 \\
> > \newline
> > \hline
> > \text{ConCerNet, $\sigma=0.05$} & 1.571 &1.319 \\
> > \newline
> > \hline
> > \text{Vanilla NN, $\sigma=0.05$} & 1.128	 & 35.161 \\
> > \newline
> > \hline
> > \text{ConCerNet, $\sigma=0.1$} & 3.677 & 5.539 \\
> > \newline
> > \hline
> > \text{Vanilla NN, $\sigma=0.1$} & 3.723 & 162.834 \\
> > \newline
> > \hline
> > \text{ConCerNet, $\sigma=1$} & 9.146 & 803.218 \\
> > \newline
> > \hline
> > \text{Vanilla NN, $\sigma=1$} & 9.610 & 1299.529 \\
> > \newline
> > \hline
> > \end{array}
> >
> > The table 2 showed the result for different noise settings for both our proposed ConCerNet and vanilla NN. Increasing noise made it difficult for both methods to learn the dynamics, while our method consistently beat the baseline by a large margin.
> >
> > 3. Regarding your suggestion on the multi-body collision example, we successfully learned the momentum conservation. First, we designed a one-dimension two body collision example, with 6 dimension system states $[m_1,x_1,v_1,m_2,x_2,v_2]$. To avoid non-continuous simulation phase transition (i.e. sudden collision), we define a permanent repulsion force (proportional to the distance) and a permanent damping force (proportional to the relative velocity). We randomly initialize the body mass, position and velocity and record the two body motions. As a result, our model could capture the conservation function proportional to $m_1v_1+m_2v_2$ which is the momentum conservation. The function has 4 inputs and as we are not allowed to post figures in openreview, we will attach the comparison figures for this example in the final version of the manuscript.

---

> > ### Author Response · Authors · 2022-11-28
> > **Follow up experiments to reviewer qJvt (2/2)**
> >
> > We hope these experiments satisfies your curiosity of the problem and convince you the power of our method. As a summary of your major concerns and our response: 1. Clarity issues: we made major modification of the writing 2. Methodology motivation: we have clarified that the simpler method by introducing prior knowledge is not a fair comparison. Even with the prior knowledge, the regularization method does not really help enforce the predicted conservation from experiment results. However, our method performs much better in conservation criterion and so is well motivated. 3. Lack of baseline: we provided HNN baseline in two of the examples, as it is not applicable to the other two.
> >
> > Regarding your suggestions on causal learning and irregular sequences, we do not think it is necessary to conduct the experiments in our paper as they are beyond the central scope. But we are happy to include additional discussions in the final version.
> >
> > From our perspective, most of your concerns are addressed in details. As we are the first work to study the conservation laws from contrastive learning perspective and propose projection method in enforcing properties in dynamics simulation, we believe both parts are of interest to the community and direct to a promising direction. We hope you can re-evaluate our paper and raise your score.
> >
> > Thanks,
> >
> > Authors

---

### Official Review · Reviewer_vHDM · 2022-10-25

**Confidence:** 3
**Correctness:** 3
**Technical Novelty And Significance:** 3
**Empirical Novelty And Significance:** 3
**Recommendation:** 6

**Clarity, Quality, Novelty And Reproducibility:**

Clarity:

The paper is well-written and easy-to-follow.

Quality:

As I mentioned in the main review, I have some concerns on (1) generalization performance, (2) hyper-parameter tuning, and (3) weak experimental results with limited comparisons.

Novely:

The idea of using the contrastive loss to model and extract invariant quantities is novel for me.

Reproducibility:

The paper contains an incomplete set of used model architectures and hyper-parameters. Code is not made publicly available.



**Strength And Weaknesses:**

Strength:
1. This paper tackles an important and challenging problem of physics (can machine learning models learn conservative dynamics and extract invariant quantities from observed trajectories without using prior knoweldge (e.g., conservation of mechanical energy) or assuming underlying structures (e.g., Hamiltonian)?).

2. To solve the above important problem, the authors introduce a constrative learning framework that can learn invariant quantities via a representation learning way. To me, this idea is novel.

3. The paper is generally easy-to-follow.

Weakness ( and Questions):
1. I am not sure whether the contrastively-learned invariant latent features, i.e., the modeled invariant quantities, can be valid for unseen initial conditions. I think the authors should report generalization performance of the proposed method to ensure it.

2. $m$, the latent feature dimension, seems to be very important hyper-parameter because it directly determines the number of invariant quantities. The authors do not explicitly state how one can tune such a important hyper-parameter when one does not know the exact value of $m$ for an unknown physical system, thus I am not sure the robustness of the proposed method in this paper.

3. Espeically, for a heat equation problem, the authors use an auto-encoder that maps an observation to $d$-dimensional laten space. It means that the modeled dynamics is $d - m$ dimensional sub-manifold and it seems that both $d$ and $m$ seem to be very sensitive hyper-parameters when learning the dynamics and invariant quantities accurately.

4. Experimental results seem to be weak. The proposed method fails to accurately learn the 4-variable Kepler system. Also, the authors only compare their model with a simple vanila network. It would be nice if the authors compared their model with some advanced models, e.g., HNN or [1].

5. Can the proposed model learn a non-conservative systems such as a dissipative one?

6. The core idea of this paper is that a certain state should have an invariant representation with respect to its time evolution. While the contrastive learning is one of good candidates to finding such a representation, there are various other methods that can enforce the invariance, e.g., consistency regularization [2]. What motivated the authors to use the contrastive learning?

***
[1] Liu, Z., & Tegmark, M. (2021). Machine learning conservation laws from trajectories. Physical Review Letters, 126(18), 180604.

[2] Sinha, S., & Dieng, A. B. (2021). Consistency regularization for variational auto-encoders. Advances in Neural Information Processing Systems, 34, 12943-12954.

**Summary Of The Paper:**

The authors propose a contrastive learning framework to model conservative dynamical systems with hidden conservative laws. More specifically, the proposed framework consists of two networks; an invariant representation network that maps elements of a certain trajectory to an identical latent features $H_{\theta_c}(x) \in \mathbb{R}^m$ which mimic $m$ invariant quantities, and a dynamics network  $\hat{f_{\theta_d}}(x) = \text{Projection}({f_{\theta_d}}(x), f: \langle f, \nabla_x H_{\theta}(x) = 0 \rangle)$ that models the targeted conservative dynamics by projecting a neural ODE vector field ${f_{\theta_d}}(x)$ to the orthogonal direction of  $\nabla_x H_{\theta}(x)$. The former and latter are trained via a neightborhood conservative loss and a standard MSE loss, respectively. The authors validate their proposed method for two simple dynamical systems (two variables with one conservative law), Kepler system (four variables with two conservative laws), and heat equation on a 1D rod (101 mesh points and one conservative law).

**Summary Of The Review:**

Although the paper is interesting, I think there is still much room for improvement, e.g., see Weakness (and Questions). My current evaluation on this paper is borderline reject.

---

> ### Author Response · Authors · 2022-11-14
> **author rebuttal (1/2)**
>
> Dear reviewer vHDM:
>
> Thank you for the feedback and pointing out your concerns, they are of great value to us, and we are very glad to clarify them in the following. As a short summary of the detailed paragraph below, we would like to clarify, 1.  We have conducted extra experiments on model generalization, parametric study on latent space. 2. Following your suggestion, we added HNN as comparison baselines in two of our experiments, as it is not applicable to two other non-Hamiltonian systems.
>
> We hope our answers address all your concerns. Please do not hesitate to contact us if you need any additional clarifications. We will do our best to resolve all your concerns.
>
>
> **$\bullet$ 1: conservation learning generalization performance** Thank you for raising the concerns about the generalization ability of our model. As we are using a general parameterization (i.e. neural networks) instead of a specific function class, like many other data-driven methods, we have little to say about the model performance on out-of-distribution testing data. We added additional testing results on out-of-distribution data for the ideal spring-mass system in Appendix A. During training, the norm of the system state ($\sqrt{x[1]^2+x[2]^2}$)is randomly sampled from $[0.3,1.2]$. Therefore, the model shows relatively good performance for states within the unit circle and cannot capture the square function for states with a norm greater than 1.5. In our paper, we aim to build a conservation field for a given data distribution and help improve the dynamics simulation within such distribution. To directly answer your question, if the unseen initial condition is drawn from similar distributions, then our model can generalize it. Otherwise, the conservation model is not valid along with the dynamics model.
>
> **$\bullet 2,3:$ autoencoder latent space and conservation function space dimension** Thank you for pointing out the concern regarding the dimension choice. To clarify your concern, we conduct a parametric study on the autoencoder space dimensions ($dim(z)$ or $d$) and conservation function dimension ($dim(H)$ or $m$) and added the results in Appendix table 5. We found the model performance is less affected by latent space dimension but drops significantly when using $m=3$. It can be explained by the unnecessary constraint limit on the dynamical model. As an application in new problems, the autoencoder dimension $d$ can be chosen by an appropriate number to represent the full system space. For conservation space $m$, we suggest users start tuning $m$ from 1 and gradually ramp up $m$ to find an optimal parameter until the model performance drops. If there exist conservation laws, using $m=1$ will always perform better than the vanilla method. If $m$ is too large, the dynamical prediction is over-constrained.
>
> **$\bullet$ 4 experiment baselines:** Thank you for pointing out the concern. For dynamical modeling baselines, we have included HNN (Ref [1]) as for the Kepler and spring mass system and append the new results in Appendix A. Its performance is similar to our methods on conservation and coordinate errors. We didn’t conduct HNN experiments on the other two examples, as it is not applicable.
> Regarding the prior work in conservation learning, we understand your concern that prior methods claim they can learn the conservation formula for systems like 4-dim Kepler system while our method failed to. However, to achieve that, it requires heavy data pre-processing and prior knowledge artificial knowledge of fitting function class to acquire the conservation laws (i.e. Ref [2]’s formula discovery is dependent on the package from Ref [3], and Ref [3] requires physics information such as symmetry and a brutal force search over combinations of given function class). Without any prior knowledge, for systems with multiple conservation laws, we can only learn the dimension of conservation space or a function of conservation functions, because any combination of conservation will be another conservation (i.e. linear combination of angular moment and total energy is also a conservation term). In our work, we proposed a generic method without any prior information and use a general neural network. Therefore, it is not appropriate to compare our methods with existing work requiring additional human knowledge. As a compromise, we can only learn the exact conservation function for systems with a single conservation law. For a system with multiple laws, our learned function can be a function of multiple laws, but the exact function is intractable. However, such information is not useless, as guaranteeing this during the prediction of the dynamic also ensures the conservation laws.

---

> > ### Author Response · Authors · 2022-11-14
> > **author rebuttal (2/2)**
> >
> > **$\bullet$ 5 dissipative system:**  Thank you for the suggestion on the dissipative system, we provided an experiment with the friction spring-mass system in Appendix B. To have the method work empirically, we need the system to be “slightly leaky” such that the conservation term dissipation within the trajectory is small compare with trajectory-wise difference. We can also introduce a ranking loss to encourage the model learns the correct sign of the dissipation function. In the dynamical system prediction, we can also guarantee the dissipation if we know the dissipation term. We can use the projection method by projecting the neural network nominal output towards the other half space instead of the conservation plane.
> >
> > **$\bullet$ 6 contrastive learning motivation:** The motivation for using contrastive learning is that it directly learns a representation that naturally clusters different state points into conserved time evolution. Thank you for suggesting consistency regularization, while we don’t agree that it can serve the same purpose, as it penalizes the sensitive perturbation of the raw data. For image classification, such penalization is understandable because a perturbed image should be classified as the original one. However, in a dynamical system with continuous labels, any perturbation does change the  system states. Therefore, it is not directly applicable to our problem.
> >
> > **$\bullet$ Reproducibility**: the experiment details are listed in Appendix A. We will surely publish the code in the final version.
> >
> > Refs:
> >
> > [1] Hamiltonian Neural Networks. Samuel Greydanus, Misko Dzamba, Jason Yosinski NeurIPS 2019
> >
> > [2] Liu, Ziming, and Max Tegmark. "Machine learning conservation laws from trajectories." Physical Review Letters 126.18 (2021): 180604.
> >
> > [3] AI Feynman: A physics-inspired method for symbolic regression. SM Udrescu, M Tegmark - Science Advances, 2020

---

> > > ### Comment · Area_Chair_v8iL · 2022-11-22
> > > **Following up on the rebuttal**
> > >
> > > Dear reviewer,
> > >
> > > the authors attempted to address your points regarding the generalization and the sensitivity of the proposed model, among other points. Are you satisfied with the response?
> > >
> > > Cheers,
> > > Your AC

---

### Official Review · Reviewer_m8A3 · 2022-10-26

**Confidence:** 4
**Correctness:** 3
**Technical Novelty And Significance:** 3
**Empirical Novelty And Significance:** 2
**Recommendation:** 3

**Clarity, Quality, Novelty And Reproducibility:**

Paper is clearly written but quality can be improved with more details in Section 3 & 4. Learning conserved quantities using contrastive learning is new.

**Strength And Weaknesses:**

Learning conservation laws from observed trajectories is an important task to study a dynamical system and can also help in generalization. To my knowledge, the contrastive learning approach is novel and interesting.



**Weaknesses**

1. The paper does not adequately showcase the utility of learning conserved quantities this way. Since conserved quantities here are learnt as a NN representation, they are not interpretable. As mentioned by the authors, the learnt quantities may be a non-linear function of the (true) relevant conserved quantities. Additionally, as seen in Figure 4, the approach is not always able to enforce the learnt conservation law properly to the forecasting network.
2. Paper does not empirically compare with any other baselines that learn conservation laws. Only comparison provided is a standard neural network which does not learn the conservation law, as expected.
3. Writing in Section 3 is rushed (possibly with mistakes) and can be improved with more details about the equations. Section 4 (and Appendix) does not provide adequate details about the experiments & training/testing methodology, and a few plots are hard to parse due to no labels.



**Questions**

1. Except the ideal spring mass experiment, every other experiment considers energy conservation where Hamiltonian/Lagrangian networks can be used, potentially with additional rotational symmetry [1, 2] for angular momentum conservation. I think these methods should be added as baselines in these experiments. Further, the paper would be much stronger if there were more experiments like spring mass experiment that conserve quantities other than energy.
2. The experiment section should include comparison with other baselines that learn conservation laws from trajectories (e.g., [3, 4] cited in the paper).
3. In Equation (2), $t_1$ is not defined, should this be $t$? Can $k = i$ in this equation?
4. In Equation (3), the expectation is defined over $x^+$ and then the summation inside is also over $x^+$. There is the same issue for $x^-$. In Equation (4), there seem to be a lot more negative samples compared to positive samples, does this not introduce some bias in learning?
5. This framework does not seem to allow for the case when two different trajectories have the same conservation value. For example, trajectories with same energy but phase-shifted. Is this true?
6. Please provide details of the experiments (in Section 4 or Appendix), for example, what are the time steps the trajectories were observed during training and the same during testing. Specifically, are the methods asked to predict beyond the times seen during training?
7. Figure 3: Please provide y-labels for all plots. In Mass/Energy conservation plots, are the true values of mass/energy learnt by the neural network H?
8. Figure 4: Please provide x-labels for plots of top row; I am assuming it is time(s) but then there is discrepancy in x-axis between top and bottom rows.
9. In Section 4.2, could you please elaborate on what type of trivial solution does the model learn. It is also not clear to me why Equation (3) will be biased toward “similarity” within trajectories, and why the proposed solution solves the issue.


**References**

[1] Finzi, Marc, et al. "Generalizing convolutional neural networks for equivariance to lie groups on arbitrary continuous data." International Conference on Machine Learning. PMLR, 2020.

[2] Finzi, Marc, Max Welling, and Andrew Gordon Wilson. "A practical method for constructing equivariant multilayer perceptrons for arbitrary matrix groups." International Conference on Machine Learning. PMLR, 2021.

[3] Ha, Seungwoong, and Hawoong Jeong. "Discovering conservation laws from trajectories via machine learning." arXiv preprint arXiv:2102.04008 (2021).

[4] Liu, Ziming, and Max Tegmark. "Machine learning conservation laws from trajectories." Physical Review Letters 126.18 (2021): 180604.

**Summary Of The Paper:**

Authors propose a contrastive learning approach to learn conserved quantities of a dynamical system using an auxiliary neural network. The neural network learns through positive samples from the same trajectory of the dynamical system and negative samples from different trajectories. The conservation is enforced during prediction by projecting the outputs from a forecasting neural network onto the conservation manifold.

**Summary Of The Review:**

My main concerns are regarding the utility of learning conservation laws this way as these NN representations are not interpretable and may not be easy to enforce onto the forecasting network. Also, paper does not adequately compare with any other baselines that learn conservation laws from trajectories.

---

> ### Author Response · Authors · 2022-11-14
> **author rebuttal (1/3)**
>
> Dear reviewer m8A3:
>
> Thank you for the feedback and pointing out your concerns, they are of great value to us, and we are very glad to clarify them in the following. As a short summary of the detailed paragraph below, we would like to clarify, 1. For single conservation law with complex terrain, our method can find the affine function of the formula. For systems with multiple conservation laws, learning the exact conservation function without prior knowledge is impossible, as any combination of conservations is also conserved. But our learned function is still useful in dynamical prediction. 2. Following your suggestion, we added HNN as comparison baselines in two of our experiments, as it is not applicable to two other non-Hamiltonian systems.
>
> We hope our answers address all your concerns. Please do not hesitate to contact us if you need any additional clarifications. We will do our best to resolve all your concerns and wish you can re-evaluate our paper.
>
>
> **$\bullet$ Weakness 1**: Thank you for mentioning the utility of learning conserved quantities. For systems with only 1 conservation law, what we learned is an affine function of the exact conservation law, which is interpretable. For the systems with multiple conservation laws, we can only learn a function of the physical conservation functions, this is not directly interpretable, but they can help encourage the conservation properties in dynamical prediction (Fig 4). There might be some misunderstanding In Figure 4, the black line is the ground truth trajectory conservation and the color lines are our prediction. Our approach can always enforce the learned conservation during prediction, but because there is an error between learned conservation and actual conservation, the prediction lines will deviate from ground truth.
>
> **$\bullet$Weakness 2**: thank you for the suggestion, we added additional results with Hamiltonian Neural Network (HNN) for comparison baselines in the modified appendix. In general, the two methods show similar performance (each method wins one experiment), and their conservation and coordinate errors are much smaller than the vanilla NN. Notice our method is applicable to general conserved dynamical system. However, HNN is only applicable to Hamiltonian systems. Therefore, we only compare the Kepler and the spring mass system.
>
> **$\bullet$Weakness 3**: thank you for pointing out the presentation issues. We polished the writing following your suggestions and update the manuscript in red. We also added the plot labels and provided more of the testing/training details in Appendix A.
>
> **Q1**: We agree with your suggestions to add more comparing methods, as we included HNN as additional experiments in the rebuttal. We guess there might be some misunderstanding in the experiments, only the spring mass system and the Kepler example can be seen as Hamiltonian dynamics and can be modeled by HNN. For other general dynamics like the chemical reaction with mass conservation and heat equation with energy conservation, HNN is not applicable.
>
> **Q2**: other conservation learning methods: Thank you for pointing it out. We didn’t include the conservation learning baseline for two reasons: 1. The existing methods heavily rely on the pre-processing and artificial knowledge on fitting function to discover the conservation laws. 2. The definition of “discovering conservation laws” is not uniform across different literature.
>
> For example, Ref [4] mostly focuses on discovering the “number” of conservation laws instead of the exact formula. Their formula recovery depends on another package called AI Feynman (Ref [5]), which needs additional case dependent knowledge (i.e. separability, symmetry, dimensions) and also case dependent fitting methods (i.e. polynomials, brute-force symbolic regression model simply tries all possible symbolic expressions within some class) to extract the formula. However, our method is fully automatic and does not assume any physics prior knowledge and formula class, therefore it is more generic and universally applicable. It is not fair to compare our neural network based model to previous methods with limited fitting formula classes and requires human knowledge in discovering process. Besides, in Ref [4] and [5], their criterion for conservation formula recovery is a boolean (“yes” or “no”), therefore, we cannot quantitatively compare with that.
>
> Ref [3] is an arxiv paper before peer review and does not include source code, so we cannot compare with it as well.
>
> **Q3**: thank you for pointing out the type. $t_1$ should be $t$ and $k$ should be $i$ in equation 2, we have marked the corrections in red in the new manuscript.

---

> > ### Author Response · Authors · 2022-11-14
> > **author rebuttal (3/3)**
> >
> > **Q9**: The trivial solution is that all the $H_\theta(x)$ outputs a similar value no matter what input $x$ is.  We observe this when the model does not have the capacity or enough data to learn a complex conservation formula even for the points on the same trajectory, and the optimization can fall into some local minimum. In Eqn (3), both minimizing the positive pair difference and enlarging the negative pair difference will decrease the loss function. If the model cannot find a constant output for all the points in the same trajectory (Kepler example, Fig 3 upper row), the model will tend to emphasize encouraging the similarity between positive pairs first, but neglects to or cannot handle the dissimilarity between negative pairs. This is what we mean by “biased toward “similarity” within trajectories”. To address that, we use the batch normalized distance instead of the absolute distance. If the model learns some trivial or similar values, the optimization can jump out of this local minimum because the relative distance instead of the absolute distance is used in the loss function.
> >
> > Refs:
> >
> > [1] Finzi, Marc, et al. "Generalizing convolutional neural networks for equivariance to lie groups on arbitrary continuous data." International Conference on Machine Learning. PMLR, 2020.
> >
> > [2] Finzi, Marc, Max Welling, and Andrew Gordon Wilson. "A practical method for constructing equivariant multilayer perceptrons for arbitrary matrix groups." International Conference on Machine Learning. PMLR, 2021.
> >
> > [3] Ha, Seungwoong, and Hawoong Jeong. "Discovering conservation laws from trajectories via machine learning." arXiv preprint arXiv:2102.04008 (2021).
> >
> > [4] Liu, Ziming, and Max Tegmark. "Machine learning conservation laws from trajectories." Physical Review Letters 126.18 (2021): 180604.
> >
> > [5] AI Feynman: A physics-inspired method for symbolic regression. SM Udrescu, M Tegmark - Science Advances, 2020
> >
> > [6] Ting Chen, Simon Kornblith, Mohammad Norouzi, and Geoffrey Hinton. A simple framework for
> > contrastive learning of visual representations. ICML 2020
> >
> > [6] Debiased Contrastive Learning. Ching-Yao Chuang, Joshua Robinson, Lin Yen-Chen Antonio Torralba, Stefanie Jegelka. NeurIPS 2020

---

> > > ### Comment · Area_Chair_v8iL · 2022-11-22
> > > **Following up**
> > >
> > > Dear reviewer,
> > >
> > > the authors tried in the rebuttal to address your points in your review. It seems you had actually similar points like reviewer kRTH. Do you agree with the argument of not comparing with previous works due to difference in starting assumptions?
> > >
> > > @Authors: I am not sure if I agree with the statement "_artificial knowledge on fitting function to discover the conservation laws_". Why is such knowledge artificial, for instance, considering that many physical laws follow an inverse squared law?
> > >
> > > Cheers,
> > > Your AC

---

> > > > ### Author Response · Authors · 2022-11-22
> > > > **Reply to AC**
> > > >
> > > > Dear AC,
> > > >
> > > > First, thank you very much for handling the paper and walking through all the reviewers' comments and our replies. We really appreciate this.
> > > >
> > > > Regarding your question, by "artificial knowledge on fitting function" we mean the two things: 1. operators requiring additional knowledge, which is different from a general fitting function (i.e. neural network). 2. knowledge on function/operator classes.
> > > >
> > > > For point 1, for instance, the inverse operation requires substantially more layers and more data to fit, comparing with a summation function. If we know such "artificial" knowledge, we can add the operators (i.e. inverse) in the neural network to make it more data efficient. We agree with your opinion that "many physical laws follow an inverse squared law" and it will be useful to add an inverse operator in our parameterization.
> > > >
> > > > For point 2, for instance, the existing work [5] limits the maximum operation numbers chosen from a given set of operations (i.e. polynomial, square root, inverse) by brute force search. This knowledge is also artificial by assuming the conservation function should have a simple format. In this case, the combination of two conservation functions will not be learned as the function is complicated.
> > > >
> > > > In our work, we do not assume any of these knowledges and use a vanilla neural network, therefore we call our method "non-artificial". We hope this clarifies your concern.
> > > >
> > > > Thanks,
> > > > Authors
> > > >
> > > >
> > > > [5] AI Feynman: A physics-inspired method for symbolic regression. SM Udrescu, M Tegmark - Science Advances, 2020

---

> > > > ### Comment · Reviewer_m8A3 · 2022-11-29
> > > > **Comparison with prior works**
> > > >
> > > > I tend to agree with the authors that it is better to require fewer assumptions or domain knowledge to learn the conservation laws. However, a comparison with these methods can still be beneficial with an emphasis that they require additional domain knowledge. Also, adding an experiment where the domain knowledge of these methods is incorrect/incomplete would improve the paper, thus showing the case when only the proposed general method will work.

---

> > > > > ### Author Response · Authors · 2022-11-29
> > > > > **Response to reviewer on Comparison with prior works**
> > > > >
> > > > > Dear reviewer:
> > > > >
> > > > > Thank you again for all the detailed response and agreement on our problem setting with fewer assumptions.
> > > > >
> > > > > Regarding the extra comparison experiments with some domain knowledge to learn the conservation function, we are considering such experiments, but probably we are not capable to pull out these experiments before the discussion window for a few reasons.  1. We are not definitive on how much "prior knowledge" to include, the obvious case is that we know exactly the conservation function class 2. the metric of "finding the conservation" is not consistent across literatures. (i.e. Ref [1] only finds the number of conservation laws and relies on Ref[2] to find the function, Ref[2] only shows qualitatively results (found or not found), which might not be applicable to our quantitative metric in Appendix D. 3. The technical difficulties to run the public code. (i.e. Ref[3] does not provide open source code. Ref[2] seems relies on specific dataset and function class, so we are not sure if we can modify it.). We hope you can understand this.
> > > > >
> > > > > However, we do provide extra experiments on modeling the dynamics with given conservation laws, as all the metrics are clear, and we have all the codes under control. With a given $H(x)$, we provide two methods to train the dynamics model: 1. using our Projection layer in NN 2. apply an additional regularization term to the loss function. We provided the results in our response to reviewer qJvt titled **Follow up experiments to reviewer qJvt (1/2)**. The results show the projection method guarantees the conservation laws and the regularization method does not help much. We believe in practice, the projection along with the known conservation laws can be a promising method and impactful in many applications.
> > > > >
> > > > > Thanks,
> > > > >
> > > > > Authors
> > > > >
> > > > > [1] Liu, Ziming, and Max Tegmark. "Machine learning conservation laws from trajectories." Physical Review Letters 126.18 (2021): 180604.
> > > > >
> > > > > [2] AI Feynman: A physics-inspired method for symbolic regression. SM Udrescu, M Tegmark - Science Advances, 2020
> > > > >
> > > > > [3] Ha, Seungwoong, and Hawoong Jeong. "Discovering conservation laws from trajectories via machine learning." arXiv preprint arXiv:2102.04008 (2021).

---

> > > ### Comment · Reviewer_m8A3 · 2022-11-29
> > > **Response to author rebuttal**
> > >
> > > Thank you for providing clarifications to my questions. However, I still have a few remaining concerns:
> > >
> > >
> > > Q1: Thank you for adding HNN to the two experiments. I may be wrong but is there a Hamiltonian or Lagrangian formulation for the heat equation (in which case HNNs or Lagrangian neural networks could be used)?
> > >
> > >
> > >
> > > Q3: I think updated Equation 2 is still incorrect: $k$ is used in LHS but not in RHS.
> > >
> > > Q4: I am still confused by the updated Equation 3. As I understand, $C$ is a trajectory sampled from distribution $D$, and $x$ is a point sampled from the trajectory $C$. $x^+$ is another point from the same trajectory that is not $x$. What is $x^+_k$?
> > >
> > >
> > >
> > > Q5: From the response, it seems to me that trajectories with the **same** value in training will create issues. I think this is a fundamental limitation of the proposed framework and not just an empirical one. For example, if the $n$ trajectories in training had $k$ different conservation values, $n>>k$, there is enough information in training to obtain the conservation law, but the proposed contrastive learning approach will not be able to learn it.
> > >
> > > I do not think the experiment with continuously generated conservation values is adequate to show this case empirically, since sampling the **same** conservation value has probability zero.
> > >
> > >
> > >
> > > Q7: The ground truth label in these plots may be misleading. I assumed ground truth meant the true mass/energy. Please clarify this in the paper.
> > >
> > >
> > >
> > > Q9: Thank you for the clarification. It would help if you could add a couple of sentences in the paper explaining this.

---

> > > > ### Comment · Reviewer_m8A3 · 2022-11-29
> > > > **Response to author rebuttal (part 2)**
> > > >
> > > > Weakness 1: I disagree with the interpretability of the learnt conservation law, even for systems with 1 conservation law. The neural network $H$ cannot be interpreted to know what the exact conservation law is, unlike prior works such as AI-Feynman.  Further, the true conservation value cannot be obtained from the learnt value $H(x)$.
> > > > A possibly better way to showcase utility of learning conservation laws in this case is by emphasizing better generalization and extrapolation to longer time horizons.
> > > >
> > > >
> > > > Weakness 2: Thank you. Please add these results to Table 1 (with N/A wherever HNN is not applicable).
> > > >
> > > >
> > > > Weakness 3: I believe there are still some issues in the equations 2 and 3 of Section 3.

---

> > > > > ### Author Response · Authors · 2022-11-29
> > > > > **Response to reviewer m8A3 part 2**
> > > > >
> > > > > Dear reviewer:
> > > > >
> > > > > Weakness 1: In our work, we claim our method can find an affine function of the exact function, as it can be seen by fig 2 and the later numerical examples in Appendix C, and we proposed a metric and experiments to show how good the learned function approximate the true conservation law in Appendix D. If the true conservation law is $f(x)=x[1]+x[2]$, then our learned function approximates $c_1*(x[1]+x[2])+c_2$, as the affine function is also conserved. This is understandable because conservation is a relative value (Ref [1] has similar argument, and they can only find an affine function of the Hamiltonian as well) and by pure data-driven method we cannot find the exact coefficient. For AI-Feynman, they include implicit prior information in fitting function classes (i.e. the coefficient must be 1 and bias constant should be 0), therefore they can find the exact function. We hope this addresses your concern on the interpretability.
> > > > >
> > > > > Regarding your suggestions on longer horizon, if the simulation state stays in the training sample distribution, the learned function is legit and the simulation will follow the conservation manifold. This is exact our motivation on improving prediction accuracy. Otherwise, we have little to say about the generalization as the neural network has never seen the out-of-distribution data (we provided the extrapolation experiments in Fig 7).
> > > > >
> > > > > Ref [1] Hamiltonian Neural Networks. Samuel Greydanus, Misko Dzamba, Jason Yosinski NeurIPS 2019
> > > > >
> > > > > Weakness 2: thank you for your suggestion on the baseline again, and we will add the results to table 1 in the main paper.
> > > > >
> > > > > Weakness 3: we reported the new modification in the prior reply.
> > > > >
> > > > > Thanks,
> > > > >
> > > > > Authors

---

> > > > ### Author Response · Authors · 2022-11-29
> > > > **Response to reviewer m8A3**
> > > >
> > > > Dear reviewer:
> > > >
> > > > Thank you for the feedback, we follow up your points in the following.
> > > >
> > > > Q1: The Hamiltonian solution does not match the heat equation, so it is not applicable. For Lagrangian system, we did a quick search through literature and found there is a solution for dissipative system in p313 in Ref [1]. While this solution requires augmenting an imaginary mirror system. In our problem setup, we do not have observation for the imaginary part and generalized momentum, so identification with LNN is not possible as well.
> > > >
> > > > Ref [1]: Methods of theoretical physics Morse, Philip M.; Feshbach, Herman.
> > > >
> > > > Q3: Sorry for the mistake, the RHS should be using $k$ instead of $j$. We will correct it.
> > > >
> > > > Q4: Sorry for the confusion again. $x^+$ denotes all the possible points from the same trajectory. $x_k^+$ denotes one of such points. For better clarification, we will delete the notation of $x^+$. In the later summation subscription, we will rewrite the summation as $\sum_{x^+_k \in C_\backslash x}$ (the openreview reply window seems not supporting multilevel of subscript, we will correct it in the paper).
> > > >
> > > > Q5: Yes, we agree with you that our continuously sampled experiment cannot prove the problem under discrete probability distribution. In our data generation process, the sampling is on continuous space, so theoretically the probability of two same conservation values is also zero. In the paper sec 3.1, we denoted **assume the initial conditions have various conservation values**, it is better to rewrite as **assume the initial conditions have varing conservation values continuously distributed on a compact set**. Regarding your concerns on the discrete conservation values, our empirical explanation is that when $k>>1$, the weight of trajectories with same conservation values is relatively small in the loss function. For extreme case when $k=2, n>>k$, it is difficult to extract the latent representation with our method as the positive pairs contaminated half of the negative labels.
> > > >
> > > > Q7: We might misunderstand your original post on Q7, we thought you were asking the relationship between exact conservation and learned $H_\theta(x)$. Please disregard our first rebuttal reply to Q7. The "ground truth" legends in the plots (fig. 3) denotes the trajectory is from data. Regarding the conservation plots (mid-column of fig 3), it means the exact conservation value of different trajectories (both data and simulation). By "exact conservation value" we mean using the known conservation function (i.e. f(x)= x[1]+x[2] for mass conservation) not the learned NN to map from states to conservation values. The ground truth you assumed is correct.
> > > >
> > > > Q9: Thank you for mentioning, we will add a few sentences in the next version.
> > > >
> > > > Thanks,
> > > >
> > > > Authors

---

> > > > ### Author Response · Authors · 2022-11-29
> > > > **Further clarification on Eqn 2**
> > > >
> > > > Dear reviewer:
> > > >
> > > > Regarding Eqn 2, we decide to replace $p(C_k|x^i_{t})$ with $p(C_i|x^i_{t})$, as we only need to consider the probability of each point belonging to its own trajectory. Following is the amended equation.
> > > >
> > > > \begin{equation}
> > > >     p(C_i|x^i_{t})\coloneqq \frac{ \sum_{t_2=1}^{T} \exp(-\lVert H_{\theta_c}(x_{t}^i)-H_{\theta_c}(x_{t_2}^i)\rVert^2)1(t_2 \neq t)}{\sum_{j=1}^{N} \sum_{t_2=1}^{T} \exp(-\lVert H_{\theta_c}(x_{t}^i)-H_{\theta_c}(x_{t_2}^j)\rVert^2)1(i\neq j \text{ or }t \neq t_2)}
> > > > \end{equation}
> > > >
> > > > We hope that clarifies.
> > > >
> > > > Thanks,
> > > >
> > > > Authors

---

> > ### Author Response · Authors · 2022-11-14
> > **author rebuttal (2/3)**
> >
> > **Q4**: thank you for noticing the bias issue. We rewrite the summation over $k$ instead of the points. Regarding the negative example bias in the loss function, this is a common issue in contrastive representation learning. The milestone contrastive learning papers (i.e. Ref [6]) manage to work with the similar bias on negative examples. Later work (Ref [7]) aims to address this issue by adding normalizing factors ahead of negative terms. In appendix D, we investigate the impact of training batch sizes (batch size = anchor trajectory 1+ negative trajectory number) towards the fitting error to the exact conservation equation. We indeed found larger batch size increases such error, and therefore we choose batch size = 10 across all conservation learning experiments. However, such bias is not a dealbreaker for the method, and we would like to investigate debiasing features in future work.
> >
> > **Q5**: We allow two trajectories to have the same or similar conservation values. In the first paragraph of section 3.1, we assume the trajectories are initialized with different conservation values, so the values are different in the training batch. The extreme case is that all the data shares the same conservation term, in that case, our method is not able to pull useful laws. Back to your question, what if we have two trajectories with same or similar values in the training batch, it will affect the model performance because similar trajectory will mislead the model. However, as long as the other trajectories in the training batch are mostly with different conservation values, it will still work. In sec 5.2, we provided a numerical experiment assuming the conservation values are continuously generated (thus allowing trajectory pair with very similar conservation value), the method still works though.
> > **Q6**: We added the extra experiment details (training/prediction trajectory length and time steps) as you requested in Appendix A Table 2 and marked it in red. For the relatively easy examples spring mass and chemical equation, the prediction time is longer than training time to reflect the error and conservation violation. For the other two cases, using short prediction time is enough to reveal the trajectory deviation from the ground truth.
> >
> > **Q7**: thank you for pointing out the missing labels. We added the missing labels (indicating the system state x[1] and x[2]) for figure 2 and 3. As we explained in Sec 4.1, the conservation is relative, not an absolute value. For the mass/energy plots, the conservation function $H_\theta()$ we learned is an affine function of the exact formula.
> >
> > **Q8**: We added the x-label (t(s)) for the upper row. The upper row is used to show the conservation learning, so the inputs are from training trajectory. The lower row is used to show the dynamics prediction, so the evaluation trajectory is used. There is no discrepancy in x-axis between top and bottom row.

---

### Official Review · Reviewer_kRTH · 2022-10-27

**Confidence:** 4
**Correctness:** 3
**Technical Novelty And Significance:** 4
**Empirical Novelty And Significance:** 2
**Recommendation:** 6

**Clarity, Quality, Novelty And Reproducibility:**

quality:

The manuscript was of high quality with no typographical errors or omissions.


clarity:

The paper was clear except for the following two points
The meaning of x∼C∼D in equation (3) was unclear.
In section 4.3, I did not understand why the reduced space was created.
If we bring all possible initial states, does the system still have 9 dimensions of freedom?


originality:

Originality is considered to be high enough to be published.


reproducibility:

Publication of the source code for the numerical experiments is desired.

**Strength And Weaknesses:**

Strengths
The framework of estimating hidden conservation laws from data and then improving simulation accuracy based on the estimated conservation laws is highly novel.


Weaknesses

A weakness of the research is that the effectiveness of the proposed method has not been verified because there is no theoretical analysis to guarantee the effectiveness of the method and the numerical experiments are limited to simple systems for which the conservation laws are known.
Concrete weaknesses are that the following two points have not been validated.


1. the effectiveness of the proposed method for learning the energy function

The denominator of the "invariant learning" loss function (Eq. (3)) seems to be based on the good properties of the energy terrain.
For example, it is questionable whether it would work effectively in the case of a complex energy function with multiple peaks, such that starting from different initial conditions would result in the same energy value.
In addition, the behavior of the model when the number of data is small relative to the degrees of freedom the system has is unclear.
Numerical experiments with more complex energy landscapes and a discussion of the relationship between the number of data and learning performance would be needed.


2. effectiveness of the proposed method as a method for estimating unknown conservation laws
The authors mention the application of the proposed method to the estimation of unknown conservation laws, but its effectiveness is questionable in terms of interpretability and feasibility of estimating complex conservation laws.


2-1. Interpretability
All of the previous studies cited in section "2.3 LEARNING WITH CONSERVED PROPERTIES" seem to achieve interpretable conservation law estimation.
On the other hand, the proposed method does not seem to be able to achieve interpretable conservation law estimation.
Methods for estimating interpretable conservation laws from DNNs trained on dynamical system data have been proposed [1,2].
I think that an additional discussion based on such previous studies seems necessary.


2-2. Possibility of Estimating Complex Conservation Laws
The authors claim that the proposed method can estimate complex conservation laws.
As an example, they demonstrate the estimation of an angular momentum conservation law for the Kepler system.
However, this task is not so difficult.
Conservation law estimation for the Kepler system has been realized in many studies [1,2].
In order to claim the effectiveness of the proposed method as a conservation law estimation method, it is necessary to at least realize conservation law estimation corresponding to symmetries for nonlinear transformations such as Runge-Lenz vectors.
Non-linear conservation estimation is also mentioned in [1][2].
Also, [3] achieves it using Hamiltonian Neural Networks by estimating the mass tensor.

[1]Ziming Liu and Max Tegmark, "Machine Learning Hidden Symmetries," PRL, 128, 180201, 2022.

[2]Yoh-ichi Mototake, "Interpretable conservation law estimation by deriving the symmetries of dynamics from trained deep neural networks," PRE, 103, 033303, 2021.

[3]Nate Gruver, Marc Finzi, Samuel Stanton, Andrew Gordon Wilson, "Deconstructing the indirect biases of hamiltonian neural networks," ICLR 2022.

**Summary Of The Paper:**

In this paper, the authors propose a method to impose an acquired conservation law on a learned time-evolving model such as a DNN.
The proposed method consists of a two-step approach. The first step is invariant learning, in which underlying conservation laws are estimated from trajectory data, and the second step is dynamic learning, in which the time evolution model is subjected to preservation of conservation laws. Numerical experiments show that the proposed method consistently outperforms the baseline method in both coordinate error and conservation measures and can be further extended to complex, large-scale dynamics by leveraging autoencoders. The proposed method may also be useful in discovering conservation laws in unknown dynamical systems.

**Summary Of The Review:**

The approach of estimating hidden conservation laws from data and improving the accuracy of simulations based on these conservation laws is an innovative research, and the novelty of this research fully satisfies the conditions for acceptance for presentation at the conference.
On the other hand, the evaluation of the effectiveness of the proposed method is weak, and the validity of the study is questionable.
I believe that additional discussion and numerical experiments to improve these points would be a condition for the conference to accept the study.

---

> ### Author Response · Authors · 2022-11-14
> **author rebuttal (1/2)**
>
> Dear reviewer kRTH:
>
> Thank you for the positive feedback and pointing out your concerns, they are of great value to us, and we are very glad to clarify them in the following. As a short summary of the detailed paragraph below, we would like to clarify, 1. For single conservation law with complex terrain, despite lacking theory, we provided many numerical experiments and another conjecture that our method can find the affine function of the formula. Following your suggestion, we conducted another numerical experiment with a conservation function with 2 peaks and our model can find it. 2. For systems with multiple conservation laws, learning the exact conservation function without prior knowledge is impossible, as any combination of conservations is also conserved. But our learned function is still useful in dynamical prediction.
>
> We hope our answers address all your concerns. Please do not hesitate to contact us if you need any additional clarifications. We will do our best to resolve all your concerns.
>
>
> **$\bullet$ 1. the effectiveness of the proposed method for learning the energy function**: Yes we agree with you that our effectiveness of the learning method is not verified by theoretical analysis, as we struggled to derive the optimal solution for Eqn (11). However, we provide a full discussion (Sec 5.2) and a conjecture (Conj. 1 in Sec 5.2) along with three numerical experiments to verify the conjecture. To address your concerns about the model capability on complex energy terrain (multiple peaks), we designed another numerical experiment in Appendix C2 with a trigonometric $g(x)$ with 2 peaks, and our optimal polynomial parameterization from contrastive learning is very close to the direct polynomial fitting to the trigonometric function. This indicates our model can handle the complex terrain you mentioned as well. As for the relationship between data size and learning performance, we provided a detailed discussion in Appendix D. A surprising result is that we found the fitting error of the learning function to the exact conservation function decrease proportionally to the data number, $\text{err} \sim \mathcal{O}(N^{-1/2})$, the rate is similar to supervised learning regression problems.
>
> **$\bullet$ 2. interpretability and feasibility of estimating complex conservation laws:**
> Thank you for raising the concern, interpretability is indeed a challenge in either conservation learning or the dynamical system modeling community. Considering the discovery process of a system with multiple conservation laws, without any physics prior information or human knowledge, we can only estimate the dimension of the conservation manifold or some function of the multiple conservation functions, because any combination of conservation functions will be another conservation function. Theoretically, it is impossible to extract the exact conservation formula for systems with multiple conservation laws just from trajectory observation. The existing works on conservation law learning require heavy data preprocessing and artificial knowledge on fitting function class to acquire the conservation laws (i.e. Ref [5] [6] requires physics information such as symmetry and a brutal force search over combinations of function class). In our work, we proposed a generic method without any prior information and use a general neural network. As a compromise, we can only learn the exact conservation function for systems with a single conservation law. For a system with multiple laws, our learned function can be a function of multiple laws, but the exact function is intractable. However, such information is not useless, as guaranteeing this during the prediction of the dynamic also ensures the conservation laws.

---

> > ### Author Response · Authors · 2022-11-14
> > **author rebuttal (2/2)**
> >
> > **$\bullet$ 2-1**: Thank you for the suggestion, we include your suggested literature for discussion Ref [1,2] in Sec 2.3.
> >
> > **$\bullet$2-2**: Thank you for mentioning the task of learning things like symmetry or Runge-Lenz vectors. As we mentioned above, without prior physics or function class information, directly extracting conservation formulas for systems with multiple conservation laws is impossible.  As the reference paper you mentioned, Ref [1] assumes knowing which symmetry property can be applied during the discovery process, and requires AI Feynman package (Ref [6]) to find certain formulas. Ref [6] is a heavy-weight package that needs prior information and traversing the possible fitting function class by brute force to find the equation. Ref [2] and Ref [3] also require the system to be Hamiltonian or satisfying other symmetries like Noether’s theorem. In general, these works limit their “learn invariants” to a certain class, therefore they can find it with prior knowledge. But their disadvantage is lacking of applicability to more general dynamics (i.e. mass conservation). Our method is more generally applicable. We hope you can understand the trade-offs of different methods, and recognize our contribution.
> >
> >
> > **$\bullet$ Eqn (3)**: Thank you for pointing out the confusion, by $x\sim C \sim D$ we meant the anchor point $x$ is sampled from a trajectory $C$ and each trajectory $C$ is drawn from a given distribution $D$. To better clarify, we rewrite the formulation as $x\sim C, C\sim D; x^+\in C_{\backslash x}; x^- \in C^-, C^- \sim D_{\backslash C}$. Other than the anchor point $x$, its positive pairs ($x^+$) are drawn from the relative complement set of $x$ in $C$. The negative pairs ($x^-$)  are drawn from other trajectories.
> >
> > **$\bullet$Sec. 4.3 reduced space**: Thank you for raising the question. We introduce the heat equation example to show that our method is generally applicable to a high-dimensional system with the help of an autoencoder and reduced space. Reduced order modeling is an efficient way to model the high-dimensional system, as similar approaches are implemented in neural network-based dynamics modeling (Ref [4]). If we directly model the large dynamical system, there requires a large model with a higher risk of overfitting. Back to the question, the heat distribution is initialized with an off-centered scaled normal distribution plus random noise, with the mean, variance and scaling factor as three random variables. However, as the PDE system is discretized into 101 nodes, the actual degree of freedom is 101. The reduced space dimension 9 is not related to the degree of freedom for the initialization or system.
> >
> > **$\bullet$Publication of the source code** We will surely publish the code in the final version.
> >
> > Refs:
> >
> > [1]Ziming Liu and Max Tegmark, "Machine Learning Hidden Symmetries," PRL, 128, 180201, 2022.
> >
> > [2]Yoh-ichi Mototake, "Interpretable conservation law estimation by deriving the symmetries of dynamics from trained deep neural networks," PRE, 103, 033303, 2021.
> >
> > [3]Nate Gruver, Marc Finzi, Samuel Stanton, Andrew Gordon Wilson, "Deconstructing the indirect biases of hamiltonian neural networks," ICLR 2022.
> >
> > [4]J Zico Kolter and Gaurav Manek. Learning stable deep dynamics models. Advances in neural
> > information processing systems, 32, 2019
> >
> > [5] Liu, Ziming, and Max Tegmark. "Machine learning conservation laws from trajectories." Physical Review Letters 126.18 (2021): 180604.
> >
> > [6] AI Feynman: A physics-inspired method for symbolic regression. SM Udrescu, M Tegmark - Science Advances, 2020

---

> > > ### Comment · Area_Chair_v8iL · 2022-11-22
> > > **Follow up**
> > >
> > > Dear reviewer,
> > >
> > > the rebuttal attempts to address the points you raised. An important point is whether the model can generalize to more complex energy terrains, and the authors have extended their setting to one with two peaks. Is this of enough complexity in your opinion? Another important point is whether the interpretability prowess of the method is of sufficient quality. The authors claim that when multiple conservation laws are preserved, it is impossible to spell out them individually without physical prior knowledge, and that previous works did so in a brute force manner. Do you agree with the statement?
> > >
> > > Cheers,
> > > Your AC

---

### Author Response · Authors · 2022-11-14
**General reply to all the reviewers**

Dear reviewers:

We would like to thank all the reviewers for their thoughtful comments. We appreciate the assessment and also value constructive suggestions from all the reviewers.

We have made significant changes in the presentation and added additional results, the modification is marked in red in the new manuscript.

For the common question, we would like to make one general reply on **experiment baselines and model capability**:

**$\bullet$** Following the reviewers’ advice, we conducted additional HNN experiments as dynamics modeling comparison baselines. It shows a similar performance to our proposed method on the two Hamiltonian systems. However, our method is more general, as it is applicable to the other two Hamiltonian systems.

**$\bullet$** For conservation learning, there are prior works claiming the discovery of conservation laws. We don’t include them as a comparison for two reasons. 1. There are different definitions of “discovering conservation laws” (some focus on the conservation dimension instead of the formula)and the criterion is not uniform (they mostly use a boolean criterion (found or not found) instead of quantitative evaluation)). 2. Almost all the prior works require heavy data pre-processing and artificial knowledge to search fitting function classes to acquire the conservation laws. Actually, for systems with more than one conservation law, it is impossible to extract the exact conservation law without prior knowledge, as any combination of invariant functions is conserved as well. Our method is more generic (no prior knowledge) and general (using NN as parameterization). As a compromise, our method only learns the exact conservation formula of systems with a single conservation law. For systems with multiple conservation laws, our model can only learn a function of the conservations but that still helps dynamical system prediction which is our major focus. We hope you can understand this scenario and recognize our contribution.

For the individual concerns/questions raised by the reviewers, we replied one by one in the following. We hope our answers address all your concerns.

Please do not hesitate to contact us if you need any additional clarifications. We will do our best to resolve all your concerns. When all your concerns are resolved, we sincerely hope that you could increase your score.

Best, Authors

---

### Decision · Program_Chairs · 2023-01-20

**Decision:**

Reject

**Justification For Why Not Higher Score:**

As written in the metareview, there should at least be more extensive experimental comparisons, and certainly one more round of reviewing given the significant changes and the back and forth with the reviewers.

**Justification For Why Not Lower Score:**

The paper is certainly interesting in its core and with a great research question.

**Metareview: Summary, Strengths And Weaknesses:**

I will start with the clear strength of this paper, which is the interesting research question it addresses. In my opinion, the right research question is fundamental to interesting and impactful research work.

The submission has led to quite some commotion and back and forth between the authors and the reviewers. However, as also indicated by the scoring, the consensus is to not accept the current work in its current form. If anything, the changes over the original manuscript are significant so that to warrant a new round of reviews from scratch. What is more, motivations and theoretical backing must be sharpened. Quoting one of the reviewers

> The correct way of learning continuous-time systems is forward integration, which derives from the very definition of the learned function, i.e., time differential. However, in this work, a gradient matching (eq.7) based approach is presented, which is how discrete-time systems are learned. The authors response to this comment was "Hamiltonian/Lagrangian NNs also perform gradient-matching". Yet, these approaches do introduce new neural net architectures instead of manifesting themselves as new dynamical learning mechanisms.

What is more, I agree that more baselines should be included. I think the authors have got this suggestion negatively. However, in my opinion this will be a strength of their paper, when they show their method can work decently enough even if no explicit prior knowledge is provided. One cannot claim generality and at the same time do not stress test the model to harder/more general settings to show the benefits. In that case, unless there is clear theoretical proof that the proposed algorithm will work, it is not possible to derive solid conclusions.

Right now, a random reader would at the very least wonder how does the method compare to other works which do have this prior knowledge, and at worse mistrust the findings due to the lack of comparisons. In a way, this is similar to how one uses the accuracy obtained by supervised training to compare with unsupervised pre-trained models on downstream tasks: one gives the supervised training results to see how far is the model from what already exists out there, with the benefit of making fewer assumptions.

All in all, I believe it is fair for the authors to go through the paper once more, and I am sure it will be read with great interest.